# ROAST: Robustifying Language Models via Adversarial Perturbation with Selective Training

**Jaehyung Kim**[†][*]  **Yuning Mao**[‡]  **Rui Hou**[‡]  **Hanchao Yu**[‡]  **Davis Liang**[‡]
**Pascale Fung**[◇]  **Qifan Wang**[‡]  **Fuli Feng**[△]  **Lifu Huang**[□]  **Madian Khabsa**[‡]

[†]KAIST,  [‡]Meta AI,  [◇]HKUST,  [△]USTC,  [□]Virginia Tech
jaehyungkim@kaist.ac.kr

## Abstract

Fine-tuning pre-trained language models (LMs) has become the *de facto* standard in many NLP tasks. Nevertheless, fine-tuned LMs are still prone to robustness issues, such as adversarial robustness and model calibration. Several perspectives of robustness for LMs have been studied independently, but lacking a unified consideration in multiple perspectives. In this paper, we propose Robustifying LMs via Adversarial perturbation with Selective Training (ROAST), a simple yet effective fine-tuning technique to enhance the multi-perspective robustness of LMs in a unified way. ROAST effectively incorporates two important sources for the model robustness, robustness on the perturbed inputs and generalizable knowledge in pre-trained LMs. To be specific, ROAST introduces adversarial perturbation during fine-tuning while the model parameters are selectively updated upon their relative importance to minimize unnecessary deviation. Under a unified evaluation of fine-tuned LMs by incorporating four representative perspectives of model robustness, we demonstrate the effectiveness of ROAST compared to state-of-the-art fine-tuning methods on six different types of LMs, which indicates its usefulness in practice.

## 1 Introduction

Fine-tuning pre-trained language models (Jing and Tian, 2020; Brown et al., 2020) has now become the *de facto* standard in many NLP tasks (Kenton and Toutanova, 2019; Liu et al., 2019; Wang et al., 2019). The typical practice for evaluating fine-tuned LMs is to measure the task performance (*e.g.,* accuracy) on a fixed (validation) set of labeled samples. However, this evaluation approach may fall short in ensuring the reliability of fine-tuned LMs, as they are still prone to *robustness* issues in real-world usage. For example, their predictions are known to be still vulnerable to small word-level

perturbations (Jin et al., 2020; Li et al., 2020), and also can be biased by the superficial cues, *e.g.*, keyword or negation (McCoy et al., 2019; Niven and Kao, 2019). As such, it is critical to additionally account for robustness during fine-tuning and evaluation of LMs.

The robustness of fine-tuned LMs has been investigated in various perspectives, such as adversarial robustness or distribution-shift generalization (Wang et al., 2021a; Zhou et al., 2021; Nam et al., 2022; Hendrycks et al., 2020). However, existing research exhibits certain limitations as these studies primarily focus on a single perspective of model robustness, rather than considering multiple perspectives simultaneously. Since real-world applications necessitate models to simultaneously exhibit robustness across multiple dimensions, a unified framework is crucial to build a reliable system. Moreover, as diverse and distinct methods are available for enhancing the robustness of each perspective, it is hard for practitioners to find an efficient approach for fine-tuning LMs to ensure such comprehensive reliability. Consequently, these limitations inspire us to explore *a single effective way for fine-tuning LMs to improve its robustness in multiple perspectives*.

In this paper, we propose a simple yet effective fine-tuning technique (Figure 1) that aims to improve the multi-perspective robustness of LMs, coined **Ro**bustifying LMs with **A**dversarial perturbation with **S**elective **T**raining (**ROAST**). The high-level idea of ROAST is effectively incorporating two important sources for the model robustness during the fine-tuning: robustness on the perturbed input and generalizable knowledge learned during pre-training of LMs. While both factors have been separately demonstrated to enhance the various facets of robustness, the collaborative framework pursuing unified robustness has been less explored yet. Specifically, ROAST first generates adversarial perturbation and adds it to the training in-

---

[*]Work done during a Meta AI internship.

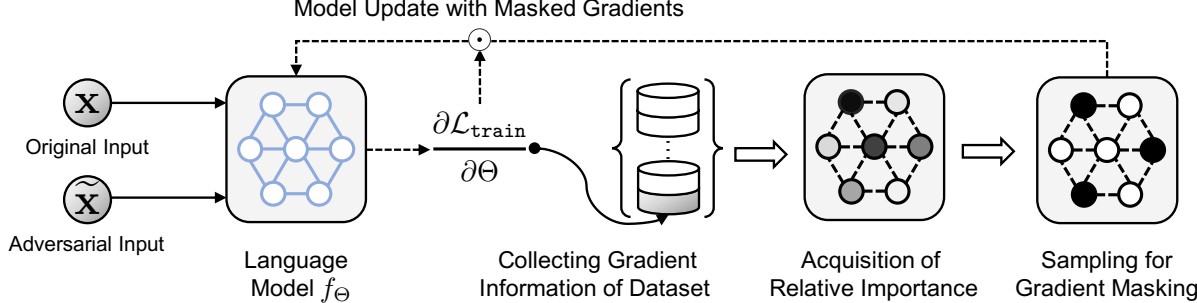

Figure 1: Illustration of ROAST: **Ro**bustifying LMs via **A**dversarial perturbation with **S**elective **T**raining.

put. Then, to prevent a large deviation from the pre-trained model while learning new task-relevant knowledge, ROAST updates model parameters *selectively*; after measuring their relative importance for solving a given task on the fly, only key responsible parameters are updated. We further justify this technique with a theoretical analysis, providing insight into the proposed selective training method.

To perform a unified evaluation of the multi-perspective robustness of LMs, we construct a new evaluation benchmark using two popular NLP tasks, *sentiment classification* and *entailment* tasks; alongside the performance on the validation set, we integrate *four distinct aspects of model robustness*: distribution-shift generalization (Ng et al., 2020), adversarial robustness (Wang et al., 2021b), model calibration (Desai and Durrett, 2020), and anomaly detection (Tack et al., 2020). Under this robustness benchmark, we demonstrate the effectiveness of ROAST for enhancing the robustness of fine-tuned LMs. Notably, across the four robustness aspects along with a standard validation accuracy, ROAST yields 18.39% and 7.63% average relative improvement compared to the traditional fine-tuning methodology on sentiment classification and entailment tasks, respectively. Furthermore, we discover that ROAST significantly improves the robustness of six state-of-the-art LMs, which further indicates its practical usefulness as a simple yet effective solution for robustiying LMs.

## 2 Background

**Multi-perspective of model robustness.** For reliable deployment in real-world scenarios (Shen et al., 2017; Gupta et al., 2021), the models should be robust in various perspectives, not just be accurate on a given validation set, drawn from the trained distribution. To this end, various perspectives of model robustness have been investigated.

*Distribution-shift (i.e., out-of-domain) generalization* evaluates how well the models can generalize to various forms of distribution shift from trained one at inference time (Ng et al., 2020; Liu et al., 2022). For example, a classifier trained on a sentiment classification dataset is expected to also perform well on another sentiment classification dataset, as both datasets are constructed to solve the same task. On the other hand, it is well known that the predictions of deep neural networks can be arbitrarily wrong, even with human-imperceptible adversarial perturbations (Goodfellow et al., 2015; Zhang et al., 2019; Wu et al., 2020). Since this problem is not exceptional for LMs with word- or sentence-level adversarial perturbation (Jin et al., 2020; Li et al., 2020), *resiliency to the adversarial examples* is also an important perspective of model robustness. In addition, *model calibration* considers the alignment between the model's predicted probability and the true correctness likelihood (Guo et al., 2017; Desai and Durrett, 2020), and hence it is essential for the reliability and interpretability of model prediction. Lastly, *anomaly detection* performance measures the model's capability to distinguish whether a given input is drawn out-of-distribution (OOD) of the training set (Hendrycks et al., 2020; Zhou et al., 2021) or not. In the vision domain, Hendrycks et al. (2022) recently investigates the effectiveness of existing data augmentation methods to enhance robustness in multiple perspectives; however, such investigation has not yet been explored in NLP.

**Enhancing robustness of language models.** To enhance the robustness of fine-tuned LMs upon (limited) training in the downstream tasks (Kim et al., 2022), various fine-tuning techniques have been explored. One line of work has explored *perturbation-based regularization*, which enhances model robustness by simulating the specific types

of perturbation to the inputs during fine-tuning. For example, Wu et al. (2021); Chen et al. (2021a) utilize Dropout to impose a stochastic perturbation to consider a different view of input, and Ng et al. (2020) substitutes the words using pre-trained masked LMs (*e.g.,* BERT (Kenton and Toutanova, 2019)). In addition, Zhu et al. (2020); Li et al. (2021) impose adversarial perturbation on word embeddings to generate a challenging view of inputs. More interestingly, Kireev et al. (2022) reveals that training with adversarial perturbation can be effective for improving model calibration. On the other hand, another line of work has focused on preserving the *generalizable knowledge* learned during pre-training of LMs; Aghajanyan et al. (2021) identifies a risk of losing the generalizable knowledge during fine-tuning and proposes a noise-based regularization to prevent it. Chen et al. (2020) directly minimizes the distance between the parameters of fine-tuned and pre-trained models as regularization, and Xu et al. (2021) proposes to only update a fixed subset of model parameters during the entire fine-tuning. A two-step strategy of linear probing and then full fine-tuning has recently been shown to be effective for distribution-shift generalization by reducing the deviation from the pre-trained model (Kumar et al., 2022). In addition, the recent result (Uppaal et al., 2023) indicates the importance of generalizable knowledge in pre-trained LMs for better anomaly detection. As both approaches enhance the different perspectives of model robustness, the effective framework for their collaborative utilization is expected to can serve as a unified way for *robusifying* LMs. However, such direction is under-explored from now on, and we try to fill this gap in this work.

## 3   Robustifying LMs via Adversarial Perturbation and Selective Training

**Overview.** In this section, we present our method, ROAST, that aims to enhance the multi-perspective robustness of LMs in a unified way. Our main idea is collaboratively incorporating two important sources of model robustness during fine-tuning; we improve the generalization of LMs on the perturbed input using *adversarial perturbation* and preserve the generalizable knowledge within pre-trained LMs via *selective training with gradient masking*. Figure 1 shows an overview of ROAST. Next, we describe in detail our techniques to improve the robustness of fine-tuned LMs.

**Adversarial training.** To improve the robustness of LM $f_\Theta$, we first incorporate adversarial perturbation during fine-tuning. Specifically, at each training iteration, we construct an adversarial example $\widetilde{\mathbf{x}}$ for training example $\mathbf{x}$. Instead of discrete token-level adversarial perturbation (Jin et al., 2020), we consider embedding-level continuous perturbation (Zhu et al., 2020) which adds noise to the word embedding of each token. Specifically, to construct $\widetilde{\mathbf{x}}$, we use a single-step gradient ascent (Jiang et al., 2020) with a step size $\delta$ under $\ell_\infty$ norm: $\widetilde{\mathbf{x}} := \mathbf{x} + \delta \cdot (\partial\mathcal{L}_{\texttt{task}}/\partial\mathbf{x})/||\partial\mathcal{L}_{\texttt{task}}/\partial\mathbf{x}||_\infty$. We then train $f_\Theta$ with a following training loss:

$$
\begin{aligned}
\mathcal{L}_{\texttt{train}} = \mathcal{L}_{\texttt{task}}(\mathbf{x}, y) + \mathcal{L}_{\texttt{task}}(\widetilde{\mathbf{x}}, y) \\
+ \lambda \mathcal{L}_{\texttt{cons}}(\mathbf{x}, \widetilde{\mathbf{x}}), \quad (1)
\end{aligned}
$$

where $\mathcal{L}_{\texttt{task}}$ is a task-specific loss (*e.g.*, cross-entropy) with given label $y$ and $\mathcal{L}_{\texttt{cons}}(\mathbf{x}, \widetilde{\mathbf{x}}) := \mathcal{D}_{\text{KL}}(f_\Theta(\mathbf{x}), f_\Theta(\widetilde{\mathbf{x}})) + \mathcal{D}_{\text{KL}}(f_\Theta(\widetilde{\mathbf{x}}), f_\Theta(\mathbf{x}))$ is bidirectional KL divergence (Wu et al., 2021).

**Selective model training via gradient masking.** However, fine-tuning with adversarial perturbation could be suboptimal in some perspectives of model robustness, as it incurs a relatively high training loss compared to naive fine-tuning (Dong et al., 2021) and hence may lose useful pre-trained knowledge due to large updates. Hence, to reduce such a potential risk, we reduce an unnecessary deviation by explicitly constraining the update. Specifically, during each iteration of the model update, we selectively update the model parameters by masking out the gradient of less important ones to solve a given task. To measure the relative importance score $s(\theta)$ of each model parameter $\theta \in \Theta$, we use the sum of the square of gradients[1]:

$$
s(\theta) = \sum_{(\mathbf{x}, y) \in \mathcal{D}} |g(\theta)|^2, \ g(\theta) = \partial\mathcal{L}_{\texttt{train}}/\partial\theta, \ (2)
$$

Note that $s(\theta)$ is highly connected to Fisher's information matrix, which provides a point estimate of the task-relevant importance of the parameters (Kirkpatrick et al., 2017; Xuhong et al., 2018). However, the calculation of gradients for all samples in $\mathcal{D}$ significantly increases training cost. To alleviate this, we approximate them by using the gradients calculated during the backward pass of model training, *i.e.*, which can be obtained *for free*. Although different values of $\theta$ are used to calculate the gradients at each training step, we empirically

---

[1] For solving the given task with $\mathcal{D}$ and training loss $\mathcal{L}_{\texttt{train}}$

verify that such an approximation is effective and remark that similar approaches have been adopted in other domains (Berthelot et al., 2019).

Using the importance scores $\{s(\theta)|\theta \in \Theta\}$, ROAST decides whether to update each model parameter $\theta$ or not. Specifically, we first obtain a normalized score $\widetilde{s}(\theta)$ from the relative order between $|s(\theta)|$ such that $\widetilde{s}(\theta_{\min}) = 0 \leq \widetilde{s}(\theta) \leq 1 = \widetilde{s}(\theta_{\max})$ where $\theta_{\min} := \arg\min_{\theta \in \Theta} s(\theta)$ and $\theta_{\max} := \arg\max_{\theta \in \Theta} s(\theta)$. Then, we set a sampling probability $p(\theta)$ using a smooth approximation of the Heaviside step function with the logistic function (Davies, 2002):

$$p(\theta) = 1 / \Big( 1 + \exp\big(2\beta(\widetilde{s}(\theta) - \alpha)\big)\Big), \quad (3)$$

where $\alpha$ and $\beta$ are hyper-parameters that control the masking ratio and smoothness, respectively. Remarkably, $p(\theta)$ becomes the hard thresholding that has been utilized in prior studies (Zhang et al., 2021; Xu et al., 2021) as $\beta \to \infty$. Compared to this, our smooth approximation enables a more calibrated use of the importance score. Then, gradient mask $m(\theta)$ is sampled from Bernoulli distribution with a probability $p(\theta)$, and ROAST selectively updates model parameters by masking the gradient using $m(\theta)$ with a scaling term $1/p(\theta)$:

$$\begin{aligned} \theta &\leftarrow \theta - \eta \cdot \widetilde{g}(\theta), \\ \widetilde{g}(\theta) &:= \big(m(\theta)/p(\theta)\big) \odot g(\theta), \end{aligned} \quad (4)$$

where $\eta$ denotes a learning rate. To demonstrate the soundness of proposed selective training with a masked gradient $\widetilde{g}(\theta)$, we derive a theoretical corollary based on previous result (Xu et al., 2021).

**Corollary.** *Under mild assumptions, the masked gradient $\widetilde{g}(\theta)$ becomes an unbiased estimator of the original gradient $g(\theta)$, i.e., $\mathbb{E}[\widetilde{g}(\theta)] = g(\theta)$, and the norm of covariance is upper bounded.*

We present a formal derivation of Corollary in Appendix B. The corollary establishes that, under certain conditions, the masked gradient $\widetilde{g}(\theta)$ is an unbiased estimator of the original gradient $g(\theta)$, which indicates that the masked gradient correctly identifies the true gradient on average; therefore, the variance introduced by masking doesn't introduce a systematic bias that could deviate the model from converging to its optimal parameters. In addition, the upper bound on the norm of covariance provides confidence in the stability of this estimator. In practical terms, it assures us that the masked gradient won't be too volatile or far off from the true gradient, thus safeguarding the convergence properties of the optimization process.

In practice, we accumulate the gradients during each training epoch to calculate the importance scores, and then generate the gradient masks for the next epoch. In addition, we empirically observe that ROAST can work without a scaling term and it sometimes outperforms the original. Hence, we consider whether to apply scaling as an additional hyper-parameter. The overall procedure of ROAST is described in Algorithm 1. More details are presented in Appendix A.3.

## 4 Experiments

In this section, we evaluate the effectiveness of ROAST to enhance the robustness of fine-tuned LMs in multi-perspectives. We first describe the experimental setups, including how the robustness evaluation benchmark is constructed, in Section 4.1. In Section 4.2, we present experimental results of ROAST and other baselines on the constructed benchmark. In Section 4.3, we provide more analysis of ROAST including (a) ablation study, (b) comparison with different methods sharing similar intuition, and (c) qualitative analysis.

### 4.1 Experimental Setup

**Tasks and metrics.** We select two popular NLP tasks, sentiment classification and entailment, to measure the robustness of fine-tuned LMs in a unified way. We take training sets of SST-2 (Socher et al., 2013) and MNLI (Williams et al., 2018) as training data for each task, respectively.
• *In-distribution performance* ($\mathrm{Acc_{in}}$): We first evaluate model performance with respect to training distribution: we measure the accuracy on the validation sets following the common practice.
• *Distribution-shift generalization* ($\mathrm{Acc_{shift}}$): To evaluate model capability on out-of-domain generalization, we measure the average accuracy on multiple distribution shifted datasets, *i.e.*, different datasets with the same task. For entailment, we follow the setups in Liu et al. (2022) – 7 different entailment datasets are used to evaluate the entailment classifier. For sentiment classification, we use 5 different datasets following Potts et al. (2021).
• *Adversarial robustness* ($\mathrm{Acc_{adv}}$): To measure the adversarial robustness of LMs, we first construct text-level adversarial examples using TextFooler (Jin et al., 2020) on vanilla fine-tuned BERT and RoBERTa models following Wang et al. (2021a).

We also consider the datasets constructed via human-in-the-loop dynamic adversarial data collection (Nie et al., 2020; Potts et al., 2021). In addition, we use the datasets from a recent benchmark for adversarial robustness, AdvGLUE (Wang et al., 2021b), which incorporate various types of adversarial noises. We report the average performance on these datasets.

• *Model calibration* (ECE): To measure the model's calibration performance, we report the average Expected Calibration Error (Guo et al., 2017), calculated during the evaluations on different datasets including in-distribution, distribution-shifted, and adversarial datasets. As a lower ECE indicates a better calibration unlike other considered robustness metrics, we denote it with (↓).

• *Anomaly detection* (AUROC): To measure model performance on anomaly detection, we use multiple external datasets as anomaly samples following the setups in recent studies (Hendrycks et al., 2020; Zhou et al., 2021). We use the maximum softmax probability score (Hendrycks et al., 2020) and AUROC for the evaluation method and metric.

To evaluate multiple robustness aspects in a unified way, we first report average relative improvement $\Delta_{\text{avg}}$ compared to the vanilla algorithm across different evaluation metrics:

$$\Delta_{\text{avg}} := \frac{1}{|\mathcal{S}|} \sum_{s \in \mathcal{S}} \Delta_s, \ \Delta_s = \frac{s - s_{\text{base}}}{s_{\text{max}} - s_{\text{base}}}, \quad (5)$$

$\mathcal{S} = \{\text{Acc}_{\text{in}}, \text{Acc}_{\text{shift}}, \text{Acc}_{\text{adv}}, \text{ECE}, \text{AUROC}\}$ and $0 \leq \Delta_s \leq 1$. $s$ and $s_{\text{base}}$ are the performances of a given method and vanilla, respectively. $s_{\text{max}}$ denotes the best score of each metric: 100 for accuracy metrics, 0 for ECE, and 1 for AUROC. Here, we note that we present the AUROC values as 100 × AUROC in our tables to maintain a consistent scale with other metrics. Also, we use a relative average instead of a direct average for measuring multi-perspective robustness, as it is more appropriate for the problem at hand; to evaluate LMs' multi-perspective robustness, it is crucial to equally incorporate multiple metrics from various perspectives. However, these metrics often have different scales, and the direct average is limited in preventing a single metric from dominating the aggregate results due to its numerical magnitude. In addition, we report the average rank, $\text{Rank}_{\text{avg}}$, among multiple baseline algorithms averaged across five different measurements in $\mathcal{S}$. More details about datasets are presented in Appendix A.1.

**Baselines.** We compare ROAST with various fine-tuning algorithms; we first consider a naïve fine-tuning method, denoted by Vanilla. We then consider a wide range of perturbation-based fine-tuning algorithms: (*1a*) WordDrop (Guo et al., 2020): dropping input words; (*1b*) HiddenCut (Chen et al., 2021a): dropping spanned hidden features; (*1c*) AdvWeight (Bahri et al., 2022): adding adversarial perturbation on the model parameters; (*1d*) AdvEmbed (Madry et al., 2018): adding adversarial perturbation on the word embeddings. (*1e*) FreeLB (Zhu et al., 2020): introducing efficient adversarial training with gradient accumulation. (*1f*) SMART (Jiang et al., 2020): in addition to adversarial perturbation, introducing EMA-based Bregman proximal point regularization. (*1g*) RIFT (Dong et al., 2021): introducing adversarial fine-tuning method from an information-theoretical perspective. Furthermore, we consider recent state-of-the-art algorithms that preserve the generalizable knowledge of pre-trained language models during fine-tuning: (*2a*) R3F (Aghajanyan et al., 2021): noise-based consistency regularization to prevent representation collapse; (*2b*) Weight Consolidation (WCons) (Chen et al., 2020): incorporation of $\ell_2$ distance between trained and pre-trained models as a regularization; (*2c*) Child-tuning (C-Tune) (Xu et al., 2021): selective update of model parameters with a sampled child model; (*2d*) LP-FT (Kumar et al., 2022): two-step strategy of linear probing and then full fine-tuning. More details of baselines are presented in Appendix A.2.

**Training details.** All models are trained using AdamW (Loshchilov and Hutter, 2019) with its default parameters $(\beta_1, \beta_2, \epsilon)$=(0.9, 0.98, 1e-6) and a weight decay of 0.01. We use linear learning rate decay with warmup ratio 0.06 and learning rate $\eta$=1e-5 (Liu et al., 2019). The models are fine-tuned with batch size 16 for 10 epochs. All the experiments are conducted with RoBERTa-large (Liu et al., 2019) except for the experiments in Table 3. Both baselines and our method are optimized with their own hyper-parameters from a set of candidates (described in Appendix A.2) based on the validation set. For ROAST, we use $\delta = 0.1$ with $\lambda \in \{0.01, 0.1, 0.5\}$ for adversarial training. For the hyper-parameters of gradient masking, we use $\alpha \in [0.6, 0.95]$, $\beta \in \{1, 5, 10\}$ along with a scaling term. More details are in Appendix A.3.

Table 1: Robustness measurements of RoBERTa-large fine-tuned on SST-2 dataset for sentiment classification. All the values and error bars are mean and standard deviation across 3 random seeds. The **best** and the second best results are in bold and underline, respectively.

| Method | $\text{Acc}_{\text{in}}$ | $\text{Acc}_{\text{shift}}$ | $\text{Acc}_{\text{adv}}$ | ECE ($\downarrow$) | AUROC | $\Delta_{\text{avg}}$ | $\text{Rank}_{\text{avg}}$ |
|---|---|---|---|---|---|---|---|
| Vanilla | $96.29_{\pm0.14}$ | $91.79_{\pm0.13}$ | $66.30_{\pm2.14}$ | $7.11_{\pm0.82}$ | $86.72_{\pm3.60}$ | 0.00 | 11.8 |
| WordDrop | $96.44_{\pm0.03}$ | $89.95_{\pm0.70}$ | $69.46_{\pm0.69}$ | $7.33_{\pm0.78}$ | $87.57_{\pm1.91}$ | -1.14 | 11.2 |
| R-Drop | $96.44_{\pm0.19}$ | $91.75_{\pm0.21}$ | $69.00_{\pm1.52}$ | $6.05_{\pm0.87}$ | $89.19_{\pm1.20}$ | 9.02 | 9.0 |
| HiddenCut | $96.67_{\pm0.34}$ | $91.14_{\pm0.20}$ | $70.32_{\pm0.78}$ | $5.47_{\pm0.43}$ | $89.91_{\pm0.38}$ | 12.26 | 5.8 |
| AdvWeight | $96.41_{\pm0.29}$ | $91.92_{\pm0.15}$ | $65.47_{\pm0.77}$ | $7.36_{\pm0.34}$ | $87.80_{\pm0.98}$ | 1.33 | 10.6 |
| AdvEmbed | $96.48_{\pm0.05}$ | $91.75_{\pm0.32}$ | $69.90_{\pm0.90}$ | $5.51_{\pm0.16}$ | $\underline{90.79_{\pm0.35}}$ | 13.70 | 6.0 |
| FreeLB | $96.33_{\pm0.34}$ | $91.94_{\pm0.38}$ | $70.06_{\pm0.27}$ | $6.49_{\pm0.51}$ | $89.82_{\pm0.40}$ | 9.21 | 7.6 |
| SMART | $\underline{96.86_{\pm0.05}}$ | $91.49_{\pm0.33}$ | $70.09_{\pm0.04}$ | $6.32_{\pm0.34}$ | $\mathbf{90.89_{\pm0.44}}$ | 13.10 | 5.8 |
| RIFT | $96.41_{\pm0.05}$ | $89.55_{\pm0.25}$ | $70.67_{\pm1.08}$ | $6.85_{\pm0.39}$ | $89.93_{\pm1.07}$ | 3.31 | 8.6 |
| ChildPTune | $96.56_{\pm0.04}$ | $91.75_{\pm0.23}$ | $69.54_{\pm0.14}$ | $5.57_{\pm0.15}$ | $87.00_{\pm0.15}$ | 7.98 | 8.2 |
| R3F | $96.56_{\pm0.09}$ | $91.79_{\pm0.09}$ | $69.13_{\pm1.55}$ | $5.83_{\pm0.15}$ | $88.49_{\pm0.49}$ | 9.38 | 7.8 |
| WCons | $96.60_{\pm0.22}$ | $\underline{92.15_{\pm0.25}}$ | $70.86_{\pm0.12}$ | $\underline{5.01_{\pm0.27}}$ | $89.61_{\pm1.03}$ | 15.47 | $\underline{3.6}$ |
| LP-FT | $96.33_{\pm0.25}$ | $91.85_{\pm0.17}$ | $\underline{72.48_{\pm0.05}}$ | $\mathbf{4.05_{\pm0.20}}$ | $89.46_{\pm0.77}$ | $\underline{16.75}$ | 5.6 |
| ROAST (Ours) | $\mathbf{96.87_{\pm0.20}}$ | $\mathbf{92.38_{\pm0.12}}$ | $\mathbf{72.57_{\pm0.53}}$ | $5.45_{\pm0.40}$ | $90.37_{\pm0.65}$ | $\mathbf{18.39}$ | $\mathbf{1.8}$ |

## 4.2 Main results

To evaluate the effectiveness of ROAST for improving the robustness of LMs, we compare it with various baselines by fine-tuning RoBERTa-large model (Liu et al., 2019). Tables 1 and 2 summarize the experimental results on sentiment classification and entailment task, respectively. First, it is worth noting that the common evaluation method using the accuracy on the validation set is not enough to capture the robustness of a given model. For example, all the baselines successfully improve the vanilla fine-tuning on the validation accuracy ($\text{Acc}_{\text{in}}$), but their robustness sometimes degrades severely as listed in Table 2. Such results support the necessity of considering multi-perspective model robustness in a unified manner, rather than naively focusing on validation accuracy. Also, we note that there is no single best method when considering multiple perspectives of model robustness; this result indicates the value of a unified measurement to facilitate robustness evaluation.

In this sense, one can observe that ROAST consistently outperforms the baseline fine-tuning methods. To be specific, across 4 different robustness metrics along with a validation accuracy, ROAST exhibits 18.39 % and 7.63% average relative improvement compared to vanilla fine-tuning on sentiment classification and entailment, respectively. As a result, ROAST achieves an average ranking of 2.7

while the previous best method achieves 4.4. These results demonstrate that ROAST could serve as a simple yet strong method for robustifying LMs. Interestingly, advanced fine-tuning methods are more effective in the sentiment classification task than in the entailment task. One possible reason is the difference in the intrinsic difficulty of the task and training dataset size; while the size of the training data is much smaller for SST-2 (67k vs MNLI: 393k), the accuracy is much higher in the sentiment classification task. This indicates that LMs could be more vulnerable to overfitting, and hence regularization of training could be more effective.

To further demonstrate the effectiveness of ROAST, we verify its compatibility across different types of pre-trained LMs. Specifically, in addition to RoBERTa-large, we conduct additional experiments with five recent state-of-the-art LMs which have the similar number of model parameters: BERT-large (Kenton and Toutanova, 2019), ALBERT-xxlarge (Lan et al., 2020), XLNet-large (Yang et al., 2019), ELECTRA-large (Clark et al., 2020), and DeBERTa-large (He et al., 2021). We note that the best hyper-parameters found in Tables 1 and 2 are inherited without additional cost from a separate search. In Table 3, we present the experimental results by comparing the average relative improvement of ROAST compared to two representative baselines, AdvEmbed and WConsol,

Table 2: Robustness measurements of RoBERTa-large fine-tuned on MNLI dataset for entailment task. All the values and error bars are mean and standard deviation across 3 random seeds. ROAST again achieves the best overall performance across different evaluation aspects.

| Method | $Acc_{in}$ | $Acc_{shift}$ | $Acc_{adv}$ | ECE ($\downarrow$) | AUROC | $\Delta_{avg}$ | $Rank_{avg}$ |
|---|---|---|---|---|---|---|---|
| Vanilla | $89.97_{\pm0.04}$ | $64.31_{\pm0.58}$ | $48.60_{\pm1.31}$ | $12.64_{\pm0.79}$ | $92.09_{\pm2.26}$ | 0.00 | 9.2 |
| WordDrop | $90.35_{\pm0.17}$ | $63.20_{\pm0.44}$ | $\underline{52.45}_{\pm0.78}$ | $12.17_{\pm0.04}$ | $87.97_{\pm1.40}$ | -8.15 | 8.2 |
| R-Drop | $\underline{90.64}_{\pm0.03}$ | $62.74_{\pm0.13}$ | $51.24_{\pm0.90}$ | $11.73_{\pm0.32}$ | $91.89_{\pm0.47}$ | 2.45 | 6.4 |
| HiddenCut | $90.48_{\pm0.18}$ | $63.84_{\pm0.26}$ | $50.43_{\pm0.12}$ | $11.83_{\pm0.26}$ | $91.64_{\pm0.39}$ | 1.60 | 6.8 |
| AdvWeight | $90.19_{\pm0.13}$ | $62.87_{\pm0.57}$ | $48.00_{\pm0.72}$ | $12.38_{\pm0.71}$ | $91.01_{\pm0.93}$ | -2.97 | 11.4 |
| AdvEmbed | $90.24_{\pm0.07}$ | $64.23_{\pm0.38}$ | $50.20_{\pm0.72}$ | $12.48_{\pm0.71}$ | $\mathbf{93.83}_{\pm0.52}$ | $\underline{5.72}$ | 6.8 |
| FreeLB | $90.27_{\pm0.10}$ | $\underline{64.94}_{\pm0.14}$ | $49.48_{\pm0.43}$ | $12.26_{\pm1.43}$ | $93.24_{\pm0.92}$ | 4.74 | 6.4 |
| SMART | $\mathbf{90.74}_{\pm0.04}$ | $63.27_{\pm0.11}$ | $52.27_{\pm0.10}$ | $11.41_{\pm0.23}$ | $91.42_{\pm0.02}$ | 2.56 | 5.8 |
| RIFT | $89.73_{\pm0.17}$ | $62.33_{\pm0.41}$ | $\mathbf{53.26}_{\pm0.97}$ | $13.46_{\pm0.35}$ | $92.88_{\pm1.54}$ | 0.85 | 9.4 |
| ChildPTune | $90.08_{\pm0.07}$ | $64.09_{\pm0.84}$ | $46.48_{\pm1.00}$ | $13.63_{\pm3.29}$ | $86.60_{\pm5.90}$ | -16.14 | 12.0 |
| R3F | $90.41_{\pm0.07}$ | $63.54_{\pm0.31}$ | $50.29_{\pm0.49}$ | $11.39_{\pm1.25}$ | $91.81_{\pm1.41}$ | 2.31 | 6.8 |
| WCons | $90.54_{\pm0.01}$ | $\mathbf{65.04}_{\pm0.55}$ | $49.00_{\pm0.10}$ | $\mathbf{10.84}_{\pm0.90}$ | $91.49_{\pm1.40}$ | 2.99 | $\underline{5.2}$ |
| LP-FT | $90.42_{\pm0.14}$ | $64.70_{\pm0.54}$ | $48.82_{\pm0.90}$ | $12.64_{\pm0.39}$ | $\underline{93.63}_{\pm0.88}$ | 5.11 | 6.6 |
| ROAST (Ours) | $\underline{90.64}_{\pm0.11}$ | $63.95_{\pm0.51}$ | $51.33_{\pm0.34}$ | $\underline{11.02}_{\pm0.45}$ | $93.25_{\pm0.15}$ | $\mathbf{7.63}$ | $\mathbf{3.6}$ |

Table 3: Robustness with different language models. Average relative improvements ($\Delta_{avg}$) compared to vanilla fine-tuned BERT-large are reported for each model. All the values are mean across 3 random seeds. More detailed experimental results, such as the performances on individual robustness metrics, are presented in Appendix D.

| | Entailment | | | | Sentiment Classification | | | |
|---|---|---|---|---|---|---|---|---|
| Model | Vanilla | AdvEmbed | WCons | ROAST | Vanilla | AdvEmbed | WCons | ROAST |
| BERT-large | 00.00 | 22.85 | 20.23 | **25.34** | 00.00 | 18.24 | 15.85 | **22.76** |
| RoBERTa-large | 37.98 | 39.35 | 40.75 | **41.31** | 38.40 | 44.94 | 47.35 | **48.33** |
| ALBERT-xxlarge | 42.86 | 42.74 | 41.43 | **44.29** | 24.75 | 45.34 | 42.36 | **51.13** |
| XLNet-large | 32.79 | 34.48 | 33.19 | **36.04** | 37.24 | 37.03 | 34.38 | **41.46** |
| ELECTRA-large | 39.47 | 41.36 | 38.49 | **42.96** | 47.85 | 48.24 | 46.21 | **49.76** |
| DeBERTa-large | 36.18 | 39.73 | 37.74 | **44.48** | 40.21 | 44.84 | 42.33 | **52.23** |

which show competitive performance in Tables 1 and 2. Here, to facilitate the comparison between different LMs, we calculate the average relative improvement using the vanilla fine-tuned BERT-large as the common baseline for $s_{base}$ in Eq. 5. We observe that ROAST significantly improves the robustness of fine-tuned models regardless of the type of LMs. More interestingly, ROAST could be useful to reveal the true potential of LMs; DeBERTa-large becomes the most robust LM with ROAST while it was originally far from the best. Detailed results on each dataset and LM are presented in Appendix D and E, respectively.

### 4.3 More analysis with ROAST

**Ablation study.** To validate the proposed components of ROAST, we conduct an ablation study; specifically, we fine-tune RoBERTa-large on SST-2 by varying each component of ROAST: (a) adversarial perturbation (*Advp*) in Eq.1 and selective update of the model parameters with (b) thresholding (*Thre*) using relative importance $s(\theta)$ in Eq.2, (c) scaling (*Scal*) with smooth approximation using $p(\theta)$ in Eq.3, and (d) sampling (*Samp*) from Bernoulli distribution with $m(\theta)$ in Eq.4. Then, we evaluate the robustness of each method using the constructed robustness benchmark as same as Table 1. We summarize the results in Table 4. As shown in Table 4, the incorporation of the proposed importance score via re-scaling, denoted by (d), performs slightly worse than naive adversarial training, denoted by (a). However, the incorporation of the importance score via a selective update with hard thresholding, denoted by (c), outperforms both.

Table 4: Ablation study with each component of RoAST on the sentiment classification with RoBERTa-large. All the values and error bars are mean and standard deviation across 3 random seeds.

| Method | AdvP | Thre | Scal | Samp | $\text{Acc}_{\text{in}}$ | $\text{Acc}_{\text{shift}}$ | $\text{Acc}_{\text{adv}}$ | ECE ($\downarrow$) | AUROC | $\Delta_{\text{avg}}$ |
|---|---|---|---|---|---|---|---|---|---|---|
| Vanilla | - | - | - | - | $96.29_{\pm0.14}$ | $91.79_{\pm0.13}$ | $66.30_{\pm2.14}$ | $7.11_{\pm0.82}$ | $86.72_{\pm3.60}$ | 0.00 |
| (a) | ✓ | - | - | - | $96.48_{\pm0.05}$ | $91.75_{\pm0.32}$ | $69.90_{\pm0.90}$ | $5.51_{\pm0.16}$ | $\underline{90.79}_{\pm0.35}$ | 13.70 |
| (b) | - | ✓ | - | - | $96.67_{\pm0.19}$ | $\underline{91.99}_{\pm0.03}$ | $70.25_{\pm0.12}$ | $5.65_{\pm0.48}$ | $89.92_{\pm0.83}$ | 13.80 |
| (c) | ✓ | ✓ | - | - | $\underline{96.71}_{\pm0.24}$ | $91.88_{\pm0.26}$ | $\underline{72.01}_{\pm0.73}$ | $\mathbf{5.14}_{\pm0.31}$ | $89.66_{\pm0.73}$ | $\underline{15.83}$ |
| (d) | ✓ | - | ✓ | - | $96.44_{\pm0.25}$ | $91.65_{\pm0.26}$ | $70.48_{\pm1.21}$ | $6.17_{\pm0.71}$ | $\mathbf{91.23}_{\pm1.94}$ | 12.28 |
| RoAST (Ours) | ✓ | ✓ | - | ✓ | $\mathbf{96.87}_{\pm0.20}$ | $\mathbf{92.38}_{\pm0.12}$ | $\mathbf{72.57}_{\pm0.53}$ | $\underline{5.45}_{\pm0.40}$ | $90.37_{\pm0.65}$ | $\mathbf{18.39}$ |

Table 5: Robustness of fine-tuned RoBERTa-large with different ways to enhance the robustness by training model under perturbation while preserving pre-trained knowledge. Rand and Min are RoAST with different masking strategies. All the values are mean across 3 random seeds and results with variance are presented in Appendix E.

| Method | Entailment | | | | | | Sentiment Classification | | | | | |
| | $\text{Acc}_{\text{in}}$ | $\text{Acc}_{\text{shift}}$ | $\text{Acc}_{\text{adv}}$ | ECE ($\downarrow$) | AUROC | $\Delta_{\text{avg}}$ | $\text{Acc}_{\text{in}}$ | $\text{Acc}_{\text{shift}}$ | $\text{Acc}_{\text{adv}}$ | ECE ($\downarrow$) | AUROC | $\Delta_{\text{avg}}$ |
|---|---|---|---|---|---|---|---|---|---|---|---|---|
| Vanilla | 89.97 | 64.31 | 48.60 | 12.64 | 92.09 | 0.00 | 96.29 | 91.79 | 66.30 | 7.11 | 86.72 | 0.00 |
| WCons* | 90.62 | 65.20 | 50.96 | 12.01 | 93.23 | 6.52 | 96.71 | 91.89 | 71.46 | 4.99 | 89.13 | 15.16 |
| LP-FT* | 90.55 | 64.88 | 48.95 | 11.46 | 92.83 | 5.29 | 96.10 | 92.14 | 70.69 | 5.18 | 90.97 | 14.24 |
| RoAST | 90.64 | 63.95 | 51.33 | 11.02 | 93.25 | 7.63 | 96.87 | 92.38 | 72.57 | 5.45 | 90.37 | 18.39 |
| Rand | 90.65 | 64.13 | 50.98 | 11.39 | 91.13 | 1.67 | 96.22 | 92.25 | 72.68 | 5.19 | 90.03 | 14.88 |
| Min | 90.68 | 64.37 | 50.87 | 11.89 | 90.89 | 0.44 | 96.22 | 91.97 | 71.45 | 5.61 | 86.70 | 7.26 |

This result indicates that the better robustness of $s(\theta)$ in Table 4 is not solely due to incorporating the importance score into the model update, but rather to the careful design of using them via selective training with sparse gradients. Moreover, RoAST explicitly constrains the model update with $m(\theta)$, significantly enhancing the robustness from its two distinct advantages for training the model. Firstly, it preserves the parameters by selectively training them using masked (*i.e.*, sparse) gradients. Secondly, it reduces distortion from the original gradients by sampling the mask instead of deterministic selection. Here, the gain from the first advantage is not only obtainable from $m(\theta)$. Hard thresholding via $s(\theta)$ also shares the same advantages of selective training and it can be verified with the results in Table 4, *i.e.*, (c) > (a), (b), (d). However, it is important to note that its effectiveness can be limited since thresholding can distort the gradients from the original direction through deterministic updates of important parameters. Therefore, the second advantage of $m(\theta)$ is crucial, as it improves selective training by reducing the risk of distortion. By sampling the mask for updates from the distribution $p(\theta)$, which prioritizes important parameters, $m(\theta)$ continuously benefits from selective training while also covering overlooked parameters through

stochastic sampling. This essential advantage of integrating $m(\theta)$ is demonstrated by improvements over (c,d) in Table 4.

**Comparison to methods with similar intuition.** The key intuition of RoAST is effectively learning a given task under challenging perturbation while preserving the generalizable knowledge within pre-trained LM. RoAST achieves this goal via selective model training, but one can consider different ways for the same purpose. For example, SMART (Jiang et al., 2020) and RIFT (Dong et al., 2021) introduce additional regularization to the preservation of pre-trained knowledge upon the adversarial training. To demonstrate the superiority of RoAST, we additionally consider the extra baselines with similar intuition, *WCons** and *LP-FT**, by incorporating adversarial perturbation to the methods for the preservation. model robust under perturbations as well. As shown in Table 1 and 2, RoAST significantly outperforms SMART and RIFT; since both approaches additionally introduce the regularization loss, it can induce the learning difficulty to find Pareto optimal between learning and preservation. Also, in Table 5, one can further verify that RoAST significantly outperforms both *WCons** and *LP-FT**. This result clearly demonstrates the effectiveness of the proposed selective

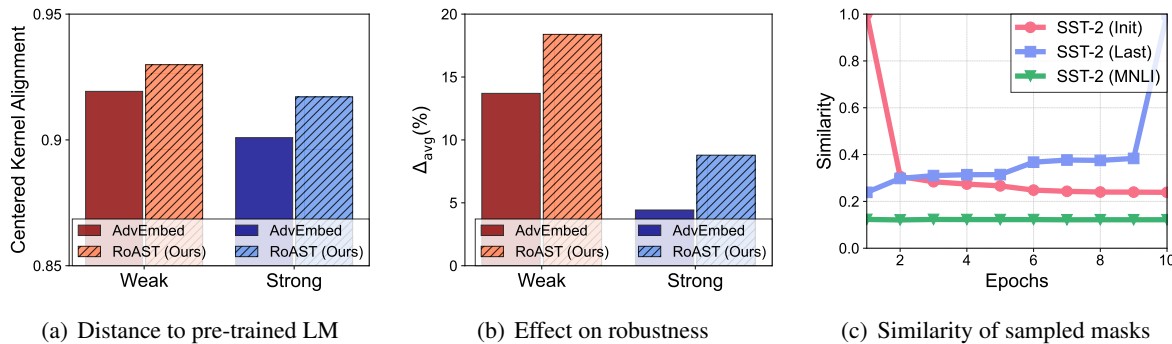

| (a) Distance to pre-trained LM | (b) Effect on robustness | (c) Similarity of sampled masks |

Figure 2: Qualitative analysis of ROAST. (a) Similarity between the initial pre-trained model and fine-tuned one. (b) Robustness under the different strengths of adversarial perturbation and improvement from ROAST. (c) Dynamics of the sampled gradient masks for selective training.

training scheme compared to naive combination with existing works.

Additionally, we conduct experiments to verify the effect of different strategies of selective training. Instead of focusing on the relatively important parameters in ROAST (Eq.3), *Min* gives more weights on the unimportant ones by considering the reversed order to obtain the normalized score, *i.e.,* $\widetilde{s}(\theta_{\max}) = 0 \leq \widetilde{s}(\theta) \leq 1 = \widetilde{s}(\theta_{\min})$. *Rand* randomly assigns $\widetilde{s}(\theta)$ regardless of the relative importance score $s(\theta)$. From Table 5, we find that the effectiveness of *Min* and *Rand* is largely degraded compared to ROAST. Remarkably, *Min* shows worse results than *Rand*, which reveals the importance of focusing on the task-relevant, important model parameters for selective training.

**Qualitative analysis.** We further present qualitative analysis on ROAST. To this end, we consider RoBERTa-large on SST-2 with two different strengths of adversarial perturbation to facilitate the analysis: *Weak* ($\lambda = 0.01$) and *Strong* ($\lambda = 0.5$). Then, we measure the similarity between the initial pre-trained model and fine-tuned one using centered kernel alignment (CKA) (Kornblith et al., 2019). Specifically, we measure the average CKA between each layer's hidden features across training epochs. In Figure 2(a), we observe that stronger adversarial training incurs larger deviation from the pre-trained model, and it could be effectively prevented by ROAST. This result indicates that ROAST helps fine-tuned model to preserve the generalizable knowledge of the pre-trained LM while learning a new task under adversarial perturbation, and hence can effectively improve the model robustness (Figure 2(b)). In addition, we investigate the dynamics of gradient masking by measuring

the intersection over union between (1) masks at the first epoch and other epochs (*SST-2 (Init)*), (2) masks at the last epoch and other epochs (*SST-2 (Last)*), and (3) masks from SST-2 and MNLI at each epoch (*SST-2 (MNLI)*). As the model is trained by focusing on the few task-relevant parameters under ROAST, the sampled masks become task-specific and far from the initial one as training goes on (Figure 2(c)). The adaptability of ROAST for each task is further observed from a low similarity between the masks of different tasks.

## 5   Conclusion

In this paper, we propose to consider multiple aspects of model robustness for the reliable deployment of LMs in real-world settings. To improve the robustness of fine-tuned LMs, we present a new fine-tuning method (ROAST) by leveraging adversarial perturbation while preventing its potential risk with efficient selective training. Through extensive experiments under constructed benchmark, we demonstrate the effectiveness of ROAST and its generalizability to multiple state-of-the-art LMs. As investing multiple aspects of model robustness in a unified viewpoint is under-explored in the literature, we expect our work to contribute to this research direction to enable us to use the well-performing pre-trained language models with more reliability. Furthermore, since our proposed method of robust fine-tuning is task- and domain-agnostic, we believe that ROAST can benefit other NLP tasks (e.g., question answering) and domains (e.g., vision and graph) as well, as interesting future work directions.

## Limitations

Although we have conducted comprehensive experiments on two representative NLP tasks with a wide range of LMs and multiple representative robustness aspects, results and analysis on more datasets, tasks, and domains (e.g., computer vision) would likely draw a more decisive conclusion. In addition to empirical evidence, theoretical analysis of why gradient masking improves language model robustness in aspects like model calibration and anomaly detection is also worth further exploration.

## Ethics Statement

The robustness issue of language models has been extensively investigated and a simple yet effective fine-tuning algorithm, ROAST, has been proposed to address this issue. Similar to other fine-tuning methods, the behavior of trained language models with the proposed method might highly rely on the training dataset. Therefore, they can suffer from the inherent problems of the given dataset, *e.g.,* gender bias (Bordia and Bowman, 2019) or annotation artifacts (Gururangan et al., 2018). However, as the proposed idea of ROAST is general and can be easily extended to other training objectives, we believe that the concerns on such problems could be alleviated by incorporating recently proposed solutions for those problems (Sagawa et al., 2020; Moon et al., 2021); for example, one could add a new training objective in Eq.1 to address each problem while preserving the idea of gradient masking of the overall objective (Eq.2).

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

# A Details on Experimental Setups

## A.1 Datasets for robustness benchmarks

As described in Section 4.1, we consider two popular NLP tasks, sentiment classification and entailment tasks, to verify the multiple aspects of model robustness with fine-tuned LMs. To fine-tune LMs for both tasks, we use SST-2 (Socher et al., 2013) and MNLI (Williams et al., 2018) datasets.

- **SST-2**: a binary single sentence classification task about movie reviews with human labels of their sentiment (*positive* or *negative*). It is composed of 67k training and 872 validation samples.

- **MNLI**: a ternary entailment task which is composed of 393k training and 20k development samples. Given a pair of sentences (premise and hypothesis), the given task is to predict whether the hypothesis is an *entailment, contradiction,* or *neutral* with respect to the premise.

Both datasets are available at https://gluebenchmark.com/tasks. After that we measure the following aspects using the described datasets.

**1.** *In-distribution accuracy* ($\text{Acc}_{\text{in}}$): To measure the performance with respect to training distribution (*i.e.,* in-distribution), we measure the accuracy on the validation sets provided from both datasets, following a usual practice (Wang et al., 2019).

**2.** *Distribution-shift generalization* ($\text{Acc}_{\text{shift}}$): To evaluate the model's capability on distribution-shift (*i.e.,* out-of-domain) generalization, we measure the accuracy on multiple distribution shifted datasets. For sentiment classification, we use the following five different sentiment datasets based on the setups in (Potts et al., 2021).

- **Yelp** (Zhang et al., 2015): The Yelp reviews dataset consists of reviews from Yelp. It is extracted from the Yelp Dataset Challenge 2015 data. As Yelp has five star-rating categories, we bin these ratings by taking the lowest two ratings to be negative and the highest two ratings to be positive, following (Potts et al., 2021). This dataset is available at https://huggingface.co/datasets/yelp_review_full.

- **Amazon** (McAuley and Leskovec, 2013): The Amazon reviews dataset consists of reviews from amazon with a rating from 1

to 5. Hence, similar to Yelp, we take review score 1 and 2 as negative, and 4 and 5 as positive. This binarized dataset is available at https://huggingface.co/datasets/amazon_polarity.

- **IMDB** (Maas et al., 2011): IMDB is a dataset for binary sentiment classification on movie reviews. It is composed of 25,000 labeled training samples and 25,000 test samples. We use IMDB dataset provided at https://huggingface.co/datasets/imdb.

- **cIMDB** (Kaushik et al., 2020): Given documents and their initial labels, (Kaushik et al., 2020) asked people to change each one so that it matched a counterfactual target label, as long as they avoided making any unnecessary changes to facts that were semantically unrelated to the label's applicability and produced revisions that resulted in internally consistent documents. The constructed cIMDB dataset is publicly available at https://github.com/acmi-lab/counterfactually-augmented-data.

- **Poem** (Sheng and Uthus, 2020): Poem is a binary single classification task about the sentiment of poem verses from Project Gutenberg. There are 892 training, 105 validation, 104 test samples, respectively. The dataset is available at https://huggingface.co/datasets/poem_sentiment

For entailment task, we follow the setups in the recent work (Liu et al., 2022); 7 different entailment datasets are used to evaluate the distribution-shift generalization of entailment classifier; here, some of the datasets are binary classification rather than ternary (denoted by $^*$). Hence, following (Liu et al., 2022), the MNLI classifier is treated as binary classifier by merging the predicts as *neutral* or *contradiction* into *not entailment*. All the datasets are available at https://github.com/alisawuffles/wanli.

- **Diag** (Wang et al., 2019): NLI Diagnostics uses naturally-occurring sentences from several domains to evaluate the variety of linguistic phenomena.

- **HANS** (McCoy et al., 2019): Based on the lexical overlap between the premise and the hypothesis, HANS seeks faulty syntactic heuristics.

- **QNLI** (Wang et al., 2019): QNLI is a binary classification task that decides whether the given (question, sentence) pair contains the correct answer (entailment) or not.

- **WNLI** (Levesque et al., 2012): Winograd NIL (WNLI) is from the Winograd Schema Challenge (Levesque et al., 2012), which checks the correct coreference through common sense. By replacing the proper referent, an entailed hypothesis is created, and by replacing the incorrect referent, a non-entailed hypothesis is created.

- **NQ-NLI** (Chen et al., 2021b): Using Natural Questions QA dataset (Kwiatkowski et al., 2019), NQ-NLI creates a decontextualized sentence from the original context for the premise and a hypothesis from a question-and-answer candidate converted into a declarative form.

- **FEVER-NLI** (Thorne et al., 2018): It is adapted from the FEVER dataset (Thorne et al., 2018). In each case, the hypothesis is a statement that is either supported (implied), refuted (contradicted), or neither (neutral), and the premise is a brief context from Wikipedia.

- **WANLI** (Liu et al., 2022): Starting with an existing dataset such as MNLI, the new samples are automatically generated with GPT-3 focusing on the ambiguous samples. To further improve the quality of constructed dataset, automatic filtering is applied and then each sample is annotated by human labelers.

3. *Adversarial robustness* ($Acc_{adv}$): To measure the adversarial robustness of model, we first construct the text-level adversarial examples using TextFooler (Jin et al., 2020) on vanilla fine-tuned BERT and RoBERT models, following (Wang et al., 2021a). We also consider the datasets constructed via dynamic adversarial data collection with human-in-the-loop (Nie et al., 2020; Potts et al., 2021). In addition, we use the datasets from a recent benchmark for adversarial robustness, AdvGLUE (Wang et al., 2021b), which incorporate the various types of adversarial noises. Overall, for the sentiment classification, the following five different adversarially constructed datasets are used to measure the adversarial robustness of fine-tuned LMs.

○ **TextFooler** (Jin et al., 2020): We denote the dataset with the adversarial texts from vanilla fine-tuned BERT as (1) *TF-B*. Similarly, the dataset with the adversarial text from vanilla fine-tuned RoBERTa is denoted as (2) *TF-R*. The official code of TextFooler is available at `https://github.com/jind11/TextFooler`.

○ **DynaSent** (Potts et al., 2021): DynaSent is dynamically constructed through multiple iterations of training a classifier model and finding its adversarial samples by involving a human annotator in the loop. In our experiments, we use the dataset from the first round, (3) *DynaSent-R1*, and the dataset from the second round, (4) *DynaSent-R2*. As DynaSent is a ternary sentiment classification (*Positive, Neutral, and Negative*), we remove the samples with Neutral. Also, we use both validation and test sets for the evaluation. The datasets are publicly released at `https://huggingface.co/datasets/dynabench/dynasent`.

○ **AdvGLUE** (Wang et al., 2021b): To construct principled and comprehensive benchmark for adversarial robustness in NLP tasks, (Wang et al., 2021b) systematically apply 14 textual adversarial attack methods to GLUE tasks to construct AdvGLUE, which is further validated by humans for reliable annotations. Hence, we measure the robustness of sentiment classifier using the dataset for SST-2 in AdvGLUE and denote it as (5) *AdvGLUE (SST-2)*. AdvGLUE dataset is available at `https://adversarialglue.github.io`.

Similarly, for entailment task, we use the following nine different adversarially constructed datasets to evaluate the adversarial robustness of entailment classifier. Here, *-m* indicates that the dataset is constructed from MNLI's matched validation set and *-mm* indicates from mismatched validation set.

- **TextFooler** (Wang et al., 2019): TF-B and TF-R are defined in the same way with the case of sentiment classification. Hence, we consider (1) *TF-B-m*, (2) *TF-B-mm*, (3) *TF-R-m*, and (4) *TF-R-mm*. We remark that the corresponding datasets constructed from other researchers are available at `https://drive.google.com/file/d/1xWwABFkzJ6fEnR1f3xr-vkMesxdO7IZm/view`.

- **ANLI** (Nie et al., 2020): Similar to the case of DynaSent, ANLI is dynamically constructed through multiple iterations of training a classifier model and finding its adversarial samples by involving a human annotator in the loop. As there are three rounds in overall, we utilize all of these datasets: (5) *ANLI-R1*, (6) *ANLI-R2*, and (7) *ANLI-R3*. ANLI dataset is available at `https://huggingface.co/datasets/anli`.

- **AdvGLUE** (Wang et al., 2021b): We use the datasets for MNLI in AdvGLUE and denote it as (8) *AdvGLUE-m* and (9) *AdvGLUE-mm*.

**4.** *Model calibration* (ECE): To measure the model's calibration performance, we report the average Expected Calibration Error (Guo et al., 2017), denoted ECE, calculated during the all evaluations on different datasets including in-distribution, distribution-shifted, and adversarial datasets introduced in above. Namely, we report the average ECE across 11 datasets for sentiment classification and 18 datasets for entailment.

**5.** *Anomaly detection* (AUROC): To measure the performance about the anomaly detection, we use the following four external datasets as anomaly samples based on the setups in the recent related works (Hendrycks et al., 2020; Zhou et al., 2021).

- □ **WMT16** (Bojar et al., 2016): WMT is a translation dataset based on the data from statmt.org and versions exist for different years using a combination of data sources. We use the English source side of English source side of English-German WMT 16, following the previous works. WMT dataset could be downloaded from `https://huggingface.co/datasets/wmt16`.

- □ **Multi30K** (Elliott et al., 2016): Multi30K is a translation datasets which extends the Flickr30K dataset with German translations created by professional translators over a subset of the English descriptions, and independently crowd-sourced descriptions of the original English descriptions. The dataset is available at `https://github.com/multi30k/dataset`.

- □ **20 NG** (Lang, 1995): 20 Newsgroup is a dataset for topic classification consists of 20 classes. 20 NG dataset is publicly available at `http://qwone.com/~jason/20Newsgroups/`.

- □ **QQP** (Sharma et al., 2019): QQP is a binary classification datasets for entailment task, where the goal is to determine if two questions in a given pair are semantically equivalent or not. As a part of GLUE benchmark, it is available at `https://huggingface.co/datasets/glue`.

In addition, we consider that the one dataset becomes anomaly samples to the other, *i.e.*, SST-2 become anomaly dataset with respect to MNLI. Hence, we use total six anomaly datasets in case of sentiment classification and five datasets in case of entailment, respectively.

## A.2 Baselines

We consider various baseline fine-tuning algorithms in NLP tasks. Specifically, we first consider a wide range of perturbation-based fine-tuning algorithms and their training loss $\mathcal{L}_{\texttt{train}}$ can be described as follow:

$$\mathcal{L}_{\texttt{train}} = \mathcal{L}_{\texttt{task}}\big(f_{\Theta}(\mathbf{x}), y\big) + \mathcal{L}_{\texttt{task}}\big(\widetilde{f}_{\Theta}(\mathbf{x}), y\big) + \lambda \mathcal{L}_{\texttt{cons}}(f_{\Theta}(\mathbf{x}), \widetilde{f}_{\Theta}(\mathbf{x})), \quad (6)$$

where $\widetilde{f}_{\Theta}(\mathbf{x})$ indicates the perturbed prediction of model $f_{\Theta}$ for input $\mathbf{x}$. Also, $\mathcal{L}_{\texttt{cons}}$ is a bidirectional KL divergence introduced in Eq.1. Here, for better explanation, we slightly abuse the notations of inputs for $\mathcal{L}_{\texttt{task}}$ and $\mathcal{L}_{\texttt{cons}}$, compared to Eq.1. With Eq.6, the baselines in this categories only have a difference in how they impose the perturbation for the prediction:

- ○ **WordDrop** (Guo et al., 2020) impose the perturbation by randomly dropping the input tokens with a probability $p_{\texttt{Wd}}$ similar to Dropout. We select $p_{\texttt{WD}} \in \{0.05, 0.10, 0.15\}$.

- ○ **HiddenCut** (Chen et al., 2021a) drops the contiguous spans within the hidden features of Transformer model during fine-tuning. As the attention-based strategy for sampling the spans shows the best results in (Chen et al., 2021a), we adopt it and tune the Hidden-Cut ratio $p_{\texttt{HC}} \in \{0.1, 0.2, 0.3\}$ with the fixed selection ratio of $0.4$. The official code is available at `https://github.com/SALT-NLP/HiddenCut`.

○ **AdvWeight** (Bahri et al., 2022) adds adversarial noise on the model parameters rather than input. It is noteworthy that this method is also called as *Sharpness-Aware Minimization (SAM)* (Foret et al., 2021). We tune the step size $\rho$ for gradient ascent step among $\{0.01, 0.03, 0.05\}$. We adopt the codes from `https://github.com/davda54/sam`.

○ **AdvEmbed** (Madry et al., 2018) imposes adversarial perturbation on the word embeddings of input tokens. We set the same values between the magnitude of perturbation and step size for the gradient ascent step; a step size $\delta$ is tuned among $\{$1e-5, 1e-3, 1e-1$\}$ under $\ell_\infty$ norm.

○ **FreeLB** (Zhu et al., 2020) proposes an efficient way to construct multi-step adversarial perturbation via gradient accumulation. We follow the best hyper-parameters provided by the authors after careful tuning. The official code is available at https://github.com/zhuchen03/FreeLB.

○ **SMART** (Jiang et al., 2020) additionally incorporates the regularization based on Breg proximal point method. Specifically, they use EMA model (Tarvainen and Valpola, 2017) and consistency loss between fine-tuned model. We set the same hyper-parameters in the paper, except the coefficient between each loss; for such coefficient, we tune among $\{$0.01, 0.1, 1.0$\}$. The official code is available at https://github.com/namisan/mt-dnn.

○ **RIFT** (Dong et al., 2021) introduce regularization loss which is derived from information-theoretical perspective. RIFT encourages an objective model to retain the features learned from the pre-trained model throughout the entire fine-tuning process. We tune hyper-parameters $\alpha$ among $\{$0.1, 0.3, 0.7$\}$, following the official code by authors: `https://github.com/dongxinshuai/RIFT-NeurIPS2021`.

Also, we commonly tune the hyper-parameter $\lambda$ among $\{0.01, 0.1, 0.5\}$ in addition to the specific hyper-parameter of each method.

On the other hand, we also consider the recent methods that prevents the model from deviating too much from the initial pre-trained model to preserve the generalizable knowledge of pre-trained language models during fine-tuning:

• **R3F** (Aghajanyan et al., 2021) introduces a noise-based consistency regularization to prevent representation collapse. Hence, we use the same candidate for $\lambda \in \{0.01, 0.1, 0.5\}$. In addition, we consider the fixed variance of noise $\sigma$=1e-5 with the two noise distributions as additional hyper-parameter $[\mathcal{U}, \mathcal{N}]$ following the original paper (Aghajanyan et al., 2021). Also, the official code is available at `https://github.com/pytorch/fairseq`.

• **Weight Consolidation** (Chen et al., 2020) incorporates $\ell_2$ distance between trained and pre-trained models as a regularization during fine-tuning. To gradually control the strength of such regularization, the authors considers the sigmoid annealing function $\lambda(t) = 1/\big(1 + \exp(-k \cdot (t - t_0))\big)$ where $t \in [0, 1]$. Here, we tune the hyper-parameters $k$ and $t_0$ among $\{0.1, 0.5, 1.0\}$ and $\{0.1, 0.3, 0.5\}$, respectively. We denote it as *WConsol*. We use the official code from `https://github.com/Sanyuan-Chen/RecAdam`.

• **Child-tuning** (Xu et al., 2021) selectively update the subset of model parameters (called child network) with a fixed child model. As the task-driven approach shows the better performance compared to task-free variant in (Xu et al., 2021), we adopt the task-driven one as baseline. We tune the child network's sparsity $p_D$ among $\{0.1, 0.2, 0.3\}$ following the original paper (Xu et al., 2021). We denote this method as *ChildTune* in our paper. Official code by the authors is publicly released at `https://github.com/PKUnlp-icler/ChildTuning`.

• **LP-FT** (Kumar et al., 2022) uses a two-step strategy of linear probing and then full fine-tuning. For a linear probing, we train the linear classifier on the frozen backbone using Adam optimizer with a fixed learning rate $\eta$ = 1e-3 and 5 epochs. Then, we tune the learning rate $\eta_{\mathtt{ft}}$ during the full fine-tuning among $\{$1e-6, 3e-5, 1e-5$\}$.

### A.3 ROAST

As described in Section 4.1, we use a fixed step size $\delta = 0.1$ for the gradient ascent step to construct

**Algorithm 1** ROAST: Robustifying LMs via Adversarial Perturbation with Selective Training

---

**Input:** Pre-trained LM $f_\Theta$, training data $\mathcal{D}$, learning rate $\eta$, update frequency $T$, masking ratio $\alpha$, smoothness factor $\beta$, adversarial noise magnitude $\delta$, coefficient of regularization $\lambda$

/* Obtaining initial gradient */
$\mathcal{G}(\Theta) \leftarrow \text{InitGrad}(f_\Theta, \mathcal{D})$
**for** each iteration t **do**
   **if** t % $T = 0$ **then**
      /* Get relative importance */
      $\{s(\theta)|\theta \in \Theta\} \leftarrow \mathcal{G}(\Theta)$, $\mathcal{G}(\Theta) \leftarrow \emptyset$
      /* Sample gradient mask */
      $\{m(\theta)|\theta \in \Theta\} \leftarrow \text{Mask}(\{s(\theta)\}, \alpha, \beta)$
   **end if**
   /* Sampling training data */
   $(\mathbf{x}, y) \sim \mathcal{D}$
   /* Add adversarially perturbation */
   $\widetilde{\mathbf{x}} \leftarrow \mathbf{x} + \delta \cdot (\partial\mathcal{L}_{\text{task}}/\partial\mathbf{x})/||\partial\mathcal{L}_{\text{task}}/\partial\mathbf{x}||_\infty$
   /* Get gradient during backward */
   $g(\theta) \leftarrow \partial\mathcal{L}_{\text{train}}/\partial\theta$, $\mathcal{L}_{\text{train}}$
   $= \mathcal{L}_{\text{task}}(\mathbf{x}, y) + \mathcal{L}_{\text{task}}(\widetilde{\mathbf{x}}, y) + \lambda\mathcal{L}_{\text{cons}}(\mathbf{x}, \widetilde{\mathbf{x}})$
   /* Update with masked gradients */
   $\theta \leftarrow \theta - \eta \cdot \big((m(\theta)/p(\theta)) \odot g(\theta)\big)$
   /* Accumulate training gradients */
   $\mathcal{G}(\theta) \leftarrow \mathcal{G}(\theta) \cup g(\theta)$
**end for**

---

the adversarial perturbation. Also, similar to the case of perturbation-based regularization methods, we tune the coefficient of consistency regularization $\mathcal{L}_{\text{cons}}$ with $\lambda \in \{0.01, 0.1, 0.5\}$ (Eq. 1). For the hyper-parameters of gradient masking, we use $\alpha \in [0.6, 0.95]$ and $\beta \in \{1, 5, 10\}$ along with the application of a scaling term. We remark that a relatively higher masking ratio $\alpha$ has been effective for sentiment classification, while the smaller $\alpha$ has been effective for entailment during our experiments. Based on such observation, we tune $\alpha$ among $\{0.95, 0.9, 0.8\}$ for sentiment classification, $\{0.6, 0.65, 0.7\}$ for entailment task along with $\beta \in \{1, 5, 10\}$. We accumulate the gradient information through each training epoch, then sample the gradient mask from them for the next epoch. In the case of InitGrad in Algorithm 1, similar to the setups in (Xu et al., 2021), we gather the sum of the square of gradients with respect to vanilla cross-entropy loss, since the classifier is not trained at that time. We use NVIDIA A100 GPUs in our experiments.

## B Proof of Corollary

In this section, we present a formal proof of the Corollary. To this end, we first present the theoretical results by (Xu et al., 2021):

**Theorem.** *(Xu et al., 2021) Suppose $\mathcal{L}$ denotes the loss function on the parameter $\theta$, for multiple data instances in the training set $\mathbf{x} \sim \mathcal{D}$, the gradients obey a Gaussian distribution $\mathcal{N}(\frac{\partial\mathcal{L}}{\partial\theta}, \sigma_g^2\mathbb{1}_k)$. For a randomly sampled batch $\mathcal{B} \sim \mathcal{S}$, when the learning algorithm is SGD with learning rate $\eta$, the probability of the gradient mask from Bernoulli distribution is p, then the mean and covariance of the update $\Delta\theta := -\eta(\frac{\partial\mathcal{L}}{\partial\theta} \odot m(\theta))$ are*

$$\mathbb{E}[\Delta\theta] = -\eta\frac{\partial\mathcal{L}}{\partial\theta},$$

$$\Sigma[\Delta\theta] = \frac{\eta^2\sigma_g^2\mathbb{1}_k}{p|\mathcal{B}|} + \frac{(1-p)\eta^2 diag\{\frac{\partial\mathcal{L}}{\partial\theta}\}^2}{p},$$

*where $\Sigma$ is the covariance matrix and $diag(X)$ is the diagonal matrix of the vector $X$.*

Here, the key difference between the above problem setup (Xu et al., 2021) and our case is the probability for masking: (Xu et al., 2021) assumes the identical Bernoulli distribution, while we assume the element-wise Bernoulli distribution with the different probability for each model parameter. However, in below Corollay, we show that our problem can be also proved in the almost same way.

**Corollary.** *We consider the same assumption in Theorem by (Xu et al., 2021), expect the probability of the gradient mask follows different Bernoulli distribution for each parameter $p(\theta)$ and $m(\theta) \sim Ber\big(p(\theta)\big)$. Then, the mean of the update $\Delta\theta$ is,*

$$\mathbb{E}[\Delta\theta] = -\eta\frac{\partial\mathcal{L}}{\partial\theta}$$

*and its covariance is bounded as,*

$$|\Sigma[\Delta\theta]||_F \leq$$

$$d\left\|\frac{\eta^2\sigma_g^2\mathbb{1}_k}{\hat{p}|\mathcal{B}|} + \frac{(1-\hat{p})\eta^2 diag\{\frac{\partial\mathcal{L}}{\partial\theta}\}^2}{\hat{p}}\right\|_F,$$

*where $\hat{p} := \min p(\theta)$.*

*Proof.* Let $g^{(i)}$ is the gradient of sample $\mathbf{x}_i$, $1 \leq i \leq |\mathcal{B}|$, then $g^{(i)} \sim \mathcal{N}(\frac{\partial\mathcal{L}}{\partial\theta}, \sigma_g^2\mathbb{1}_k)$ by the assumption. Let $g = \sum_{i=1}^{|\mathcal{B}|} \frac{g_i}{|\mathcal{B}|}$, then we have

$$\Delta\theta = -\eta\big(\sum_{i=1}^{|\mathcal{B}|}\frac{g^{(i)}}{|\mathcal{B}|}\big) \odot m(\theta) = -\eta g \odot m(\theta)$$

When we consider $g$, the followings are obtained:

$$\mathbb{E}[g] = \frac{\partial \mathcal{L}}{\partial \theta}, \Sigma[g] = \frac{\sigma_g^2 \mathbb{1}_k}{|\mathcal{B}|}$$

Suppose $\widetilde{g} := \big(m(\theta)/p(\theta)\big) \odot g$, then we have:

$$\mathbb{E}[\widetilde{g}] = \frac{p(\theta)}{p(\theta)} \times \frac{\partial \mathcal{L}}{\partial \theta} = \frac{\partial \mathcal{L}}{\partial \theta} = \mathbb{E}[g]$$

Let $\widetilde{g}_i, g_i, p_i$ are the $i$-th dimension of $\widetilde{g}, g, p$. Then,

$$
\begin{aligned}
Var[\widetilde{g}_i] &= \mathbb{E}[\widetilde{g}_i^2] - (\mathbb{E}[\widetilde{g}_i])^2 \\
&= p_i \mathbb{E}[(\frac{g_i}{p_i})^2] - (\mathbb{E}[\widetilde{g}_i])^2 \\
&= \frac{\mathbb{E}[g_i^2]}{p_i} - (\mathbb{E}[\widetilde{g}_i])^2 \\
&= \frac{(\mathbb{E}[g_i])^2 + Var[g_i]}{p_i} - (\mathbb{E}[\widetilde{g}_i])^2 \\
&= \frac{Var[g_i]}{p_i} + \frac{(1-p_i)(\mathbb{E}[\widetilde{g}_i])^2}{p_i}
\end{aligned}
$$

Since $\frac{1}{p_i}$ is a decreasing function regarding $p_i$, one can derive following bound with $\hat{p} := \min_i p_i$,

$$||\Sigma[\widetilde{g}]||_F \leq \left\| \frac{\sigma_g^2 \mathbb{1}_k}{\hat{p}|\mathcal{B}|} + \frac{(1-\hat{p})\text{diag}\{\frac{\partial \mathcal{L}}{\partial \theta}\}^2}{\hat{p}} \right\|_F ,$$

# C  More Quantitative Results with RoAST

## C.1  Generalization beyond embedding-level perturbation

For RoAST, we chose embedding-level perturbation (Zhu et al., 2020; Jiang et al., 2020) over token-level perturbation (Jin et al., 2020; Li et al., 2020), as it is more computationally efficient and better suited for enhancing multi-perspective robustness. Specifically, the construction of token-level adversarial perturbations requires more computation to solve the discrete optimization problem, and additional regularization is often introduced to prevent degenerate cases such as significant changes in semantic or lexical violation (Jin et al., 2020; Li et al., 2020; Park et al., 2022). For example, a relatively simple construction of token-level perturbation with Park et al. (2022) requires 15% more times per iteration, compared to embedding-level perturbation. In addition, while the token-level adversarial perturbation is effective for adversarial robustness, it often comes at the cost of a decrease

in clean accuracy (Dong et al., 2021) due to the relatively large perturbation on a discrete space. In contrast, the embedding-level perturbation can be constructed by adding small noise to continuous space, making it more feasible to train the model without the loss of accuracy (Zhu et al., 2020; Jiang et al., 2020)

To further validate the generalization of RoAST beyond the embedding-level perturbation, we conduct additional experiments by adapting RoAST with discrete token-level adversarial perturbations by fine-tuning RoBERTa-large using VAT-D (Park et al., 2022). We used the same hyper-parameters for discrete token-level perturbation as in Park et al. (2022), and for RoAST, we used the same values previously found with embedding-level perturbation. The results are shown in the table below.

Here, we observe that the discrete token-level adversarial perturbation can improve the multi-perspective robustness, especially for adversarial robustness ($\text{Acc}_{\text{adv}}$). However, it comes at the cost of degradation in clean accuracy ($\text{Acc}_{\text{in}}$). With RoAST, such degradation can be mitigated and the overall robustness of the model could be improved.

## C.2  Absolute average improvement of RoAST

During the experiments, we used the average of relative improvement, instead of absolute improvement, as it is more appropriate to measure the multi-perspective robustness than the average of absolute improvement, not to scale up the values. However, we also recognize its weak points, such as the risk of amplification, which is the reason why we additionally report $\text{Rank}_{\text{avg}}$, which does not have similar issues. We emphasize that RoAST exhibits the lowest rank among the state-of-the-art fine-tuning methods on both sentiment classification and entailment tasks. Nevertheless, to address the concerns about this, we additionally calculate the absolute average improvement ($\text{Abs}_{\text{avg}}$) of {$\text{Acc}_{\text{in}}$, $\text{Acc}_{\text{shift}}$, $\text{Acc}_{\text{adv}}$, 100 - ECE, 100 * AUROC}, on the sentiment classification task. The results are presented in Table 7. Here, one can observe that our method still outperforms the state-of-the-art fine-tuning method with a large gap; RoAST exhibits 46.72% relative improvement on $\text{Abs}_{\text{avg}}$, compared to SMART (1.16% vs 0.79%). This result further demonstrates the effectiveness of our method, and we do believe that our empirical results clearly show the merit of the proposed framework.

Table 6: Generalization of ROAST with discrete token-level adversarial perturbation. Here, for such perturbation, we use VAT-D (Park et al., 2022). All the values are mean across 3 random seeds.

| Method | $\text{Acc}_{\text{in}}$ | $\text{Acc}_{\text{shift}}$ | $\text{Acc}_{\text{adv}}$ | ECE ($\downarrow$) | AUROC | $\Delta_{\text{avg}}$ |
|---|---|---|---|---|---|---|
| Vanilla | 96.29 | 91.79 | 66.30 | 7.11 | 86.72 | 0.00 |
| VAT-D | 95.83 | 90.86 | 70.37 | 6.33 | 90.39 | 5.37 |
| VAT-D + ROAST | 96.27 | 90.45 | 72.12 | 6.94 | 92.16 | 8.73 |

Table 7: Average absolute improvements with different baselines. All the values are mean across 3 random seeds.

| Method | Vanilla | WordDrop | R-Drop | HiddenCut | AdvWeight | AdvEmbed | FreeLB | SMART | RIFT | ChildPTune | R3F | WCons | LP-FT | RoAST (Ours) |
|---|---|---|---|---|---|---|---|---|---|---|---|---|---|---|
| $\text{Abs}_{\text{avg}}$ | 76.47 | 76.36 | 76.96 | 76.91 | 76.94 | 77.20 | 77.13 | 77.26 | 76.95 | 74.72 | 76.93 | 77.05 | 76.99 | **77.63** |

## C.3 Additional comparison with FreeLB++

Here, we present additional experimental results with FreeLB++ (Li et al., 2021) on the sentiment classification task, which runs adversarial perturbation for 10 steps without constraints of norm-bounded projection. We also tuned the adversarial step size appropriately as the number of steps varies. The results are summarized in Table 8. Here, one can observe that FreeLB++ outperforms FreeLB, especially in $\text{ACC}_{\text{adv}}$ (70.07 $\rightarrow$ 72.45). Consequently, FreeLB++ is better than FreeLB for multi-perspective robustness as well ($\Delta_{\text{avg}}$ : 9.21 $\rightarrow$ 11.02). However, RoAST still outperforms FreeLB++ with a large gap, which further demonstrates the effectiveness of our method. On the other hand, these results indicate a potential room for further improvement in our method at the additional cost of the increased number of steps, since RoAST also uses a single step for constructing adversarial perturbation.

## C.4 Relationship between overfitting and robustness

To investigate the relationship between overfitting and robustness of LMs, we performed additional experiments by training RoBERTa-large on SST-2, using the Vanilla method with a constant learning rate for 100 epochs. As shown in Table 9, the model's robustness significantly decreases as the training progresses. This outcome confirms that preserving the parameters is critical for maintaining model robustness, which is a fundamental principle of selective training with $m(\theta)$ in ROAST.

## D Additional Results with Various LMs

First, we present the detailed results on the individual robustness metrics like Tables 1 and 2. In Table 10 and 11, the results on sentiment classification and entailment are presented, respectively.

Next, we provide the additional results with different vanilla algorithm to calculate $\Delta_{\text{avg}}$; In Table 3, we use BERT-large as a universal vanilla algorithm to calculate relative improvement across different LMs to facilitate comparison in terms of multi-perspective robustness. This choice was made to provide additional insight into the question of *"which LM is the most robust"*. While answering this question is important, we also acknowledge that using the corresponding LM's score as the baseline could provide additional insights into how RoAST performs with each specific LM. Hence, we recalculate Table 3 with the corresponding LM's score and present it in Table 12. One can observe that ROAST mostly outperforms the baselines except in only 1 case, and achieves large improvements compared to baselines in both tasks. This result demonstrates the robustness and effectiveness of RoAST with respect to different LMs.

Lastly, we conducted additional experiments on the sentiment classification task with GPT2-large (Radford et al., 2019), a popular decoder-only LM, to validate the applicability of our approach. Following Radford et al. (2019), we added a linear classifier head on the last token's embedding output for fine-tuning. The results are presented in Table 13. Here, we first observe that the improvements with baseline methods are largely limited. We speculate that this ineffectiveness may occur due to the different nature of decoder-only LMs compared to encoder-only ones, such as BERT, as it could result in different effectiveness of the baseline algorithms and tuned hyper-parameters, which were originally developed and demonstrated only using encoder-only models; for instance, most of the baselines (Jiang et al., 2020; Zhu et al., 2020; Chen et al.,

Table 8: Comparison with additional baseline, FreeLB++ (Li et al., 2021) on the sentiment classification task. All the values are mean across 3 random seeds.

| Method | $\mathrm{Acc_{in}}$ | $\mathrm{Acc_{shift}}$ | $\mathrm{Acc_{adv}}$ | ECE ($\downarrow$) | AUROC | $\Delta_{\mathrm{avg}}$ |
|---|---|---|---|---|---|---|
| Vanilla | 96.29 | 91.79 | 66.30 | 7.11 | 86.72 | 0.00 |
| FreeLB | 96.33 | 91.94 | 70.07 | 6.49 | 89.82 | 9.21 |
| FreeLB++ | 96.14 | 91.79 | 72.45 | **5.44** | **90.79** | 11.02 |
| RoAST (Ours) | **96.87** | **92.38** | **72.57** | 5.45 | 90.37 | **18.39** |

Table 9: Tradeoff between robustness and training epochs. All the values are mean across 3 random seeds.

| Method | $\mathrm{Acc_{in}}$ | $\mathrm{Acc_{shift}}$ | $\mathrm{Acc_{adv}}$ | ECE ($\downarrow$) | AUROC | $\Delta_{\mathrm{avg}}$ |
|---|---|---|---|---|---|---|
| Vanilla (Orig) | 96.29 | 91.79 | 66.30 | 7.11 | 86.72 | 0.00 |
| Epoch 20 | 94.88 | 90.66 | 64.82 | 7.75 | 83.83 | -17.45 |
| Epoch 40 | 93.92 | 88.52 | 60.54 | 9.81 | 77.04 | -46.38 |
| Epoch 60 | 93.42 | 88.11 | 59.63 | 10.16 | 72.28 | -58.74 |
| Epoch 80 | 93.39 | 87.43 | 59.03 | 10.68 | 73.83 | -60.08 |
| Epoch 100 | 93.31 | 87.52 | 58.28 | 10.62 | 67.61 | -69.94 |

2020; Xu et al., 2021) have been demonstrated under encoder-only models and not shown the results of decoder-only ones. Nevertheless, our RoAST approach continues to enhance the multi-perspective robustness of GPT2-large, with an average relative improvement of 5.07% compared to the Vanilla method. These results indicate that the effectiveness of our approach is not limited to BERT-based LMs with Transformer-encoder architecture.

# E  Individual Experimental Results

Next, we present the results on each dataset for each robustness metric. First, we present the results from sentiment classification. Specifically, we report the accuracy and ECE on distribution shifted datasets in Table 14 and 15, respectively. Also, we report the accuracy and ECE on adversarially constructed datasets in Table 16 and 17, respectively. The results of anomaly detection are shown in Table 18. Next, we present the results from entailment task; we report the accuracy and ECE on distribution shifted datasets in Table 19 and 20, respectively. Then, we report the accuracy and ECE on adversarially constructed datasets in Table 21 and 22, respectively. Finally, we report the results of anomaly detection in Table 23.

Table 10: Robustness measurements of six different LMs fine-tuned on SST-2 dataset for sentiment classification. All the values and error bars are mean and standard deviation across 3 random seeds.

| Type of LM | Method | $Acc_{in}$ | $Acc_{shift}$ | $Acc_{adv}$ | ECE ($\downarrow$) | AUROC |
|---|---|---|---|---|---|---|
| BERT | Vanilla | $94.04_{\pm0.28}$ | $89.30_{\pm0.54}$ | $57.10_{\pm1.65}$ | $9.21_{\pm2.01}$ | $83.68_{\pm0.69}$ |
| | AdvEmbed | $94.42_{\pm0.14}$ | $88.75_{\pm0.75}$ | $61.16_{\pm0.26}$ | $9.46_{\pm1.67}$ | $83.40_{\pm1.13}$ |
| | WConsol | $93.81_{\pm0.16}$ | $89.46_{\pm0.61}$ | $56.34_{\pm1.63}$ | $9.35_{\pm0.65}$ | $85.09_{\pm0.35}$ |
| | RoAST (Ours) | $94.07_{\pm0.20}$ | $88.87_{\pm0.31}$ | $60.31_{\pm0.53}$ | $7.00_{\pm0.49}$ | $85.89_{\pm0.29}$ |
| RoBERTa | Vanilla | $96.29_{\pm0.14}$ | $91.79_{\pm0.13}$ | $66.30_{\pm2.14}$ | $7.11_{\pm0.82}$ | $86.72_{\pm3.60}$ |
| | AdvEmbed | $96.48_{\pm0.05}$ | $91.75_{\pm0.32}$ | $69.90_{\pm0.90}$ | $5.51_{\pm0.16}$ | $90.79_{\pm0.35}$ |
| | WConsol | $96.60_{\pm0.22}$ | $92.15_{\pm0.25}$ | $70.86_{\pm0.12}$ | $5.01_{\pm0.27}$ | $89.61_{\pm1.03}$ |
| | RoAST (Ours) | $96.87_{\pm0.20}$ | $92.38_{\pm0.12}$ | $72.57_{\pm0.53}$ | $5.45_{\pm0.40}$ | $90.37_{\pm0.65}$ |
| ALBERT | Vanilla | $95.24_{\pm0.52}$ | $88.63_{\pm0.59}$ | $64.13_{\pm0.31}$ | $8.71_{\pm2.34}$ | $88.15_{\pm1.30}$ |
| | AdvEmbed | $96.52_{\pm0.39}$ | $91.78_{\pm0.17}$ | $73.63_{\pm1.69}$ | $5.88_{\pm0.30}$ | $87.34_{\pm0.54}$ |
| | WConsol | $96.25_{\pm0.66}$ | $90.92_{\pm1.20}$ | $71.95_{\pm4.64}$ | $5.75_{\pm1.07}$ | $87.44_{\pm2.27}$ |
| | RoAST (Ours) | $96.62_{\pm0.06}$ | $92.33_{\pm0.24}$ | $78.20_{\pm0.64}$ | $5.00_{\pm0.10}$ | $89.32_{\pm0.20}$ |
| XLNet | Vanilla | $95.87_{\pm0.34}$ | $90.80_{\pm0.31}$ | $68.33_{\pm0.83}$ | $6.57_{\pm0.46}$ | $86.76_{\pm2.47}$ |
| | AdvEmbed | $96.25_{\pm0.29}$ | $91.22_{\pm0.61}$ | $67.39_{\pm1.42}$ | $7.56_{\pm1.59}$ | $88.26_{\pm2.82}$ |
| | WConsol | $95.26_{\pm0.35}$ | $90.60_{\pm0.51}$ | $67.77_{\pm1.70}$ | $6.82_{\pm0.84}$ | $88.58_{\pm3.53}$ |
| | RoAST (Ours) | $96.02_{\pm0.30}$ | $90.61_{\pm0.46}$ | $69.61_{\pm1.29}$ | $5.48_{\pm0.62}$ | $92.25_{\pm0.30}$ |
| ELECTRA | Vanilla | $96.96_{\pm0.06}$ | $91.62_{\pm0.10}$ | $74.15_{\pm0.27}$ | $4.92_{\pm0.07}$ | $82.43_{\pm1.31}$ |
| | AdvEmbed | $97.13_{\pm0.08}$ | $90.97_{\pm0.46}$ | $74.81_{\pm0.16}$ | $5.20_{\pm1.78}$ | $88.98_{\pm1.07}$ |
| | WConsol | $97.08_{\pm0.06}$ | $91.51_{\pm0.24}$ | $73.16_{\pm0.69}$ | $5.55_{\pm0.08}$ | $82.40_{\pm1.82}$ |
| | RoAST (Ours) | $97.02_{\pm0.19}$ | $91.75_{\pm0.08}$ | $74.74_{\pm3.41}$ | $4.97_{\pm0.19}$ | $88.87_{\pm0.67}$ |
| DeBERTa | Vanilla | $96.48_{\pm0.20}$ | $90.95_{\pm0.74}$ | $69.96_{\pm1.25}$ | $6.63_{\pm0.69}$ | $86.72_{\pm4.27}$ |
| | AdvEmbed | $96.64_{\pm0.05}$ | $91.83_{\pm0.44}$ | $71.18_{\pm0.50}$ | $6.08_{\pm5.72}$ | $90.21_{\pm0.86}$ |
| | WConsol | $96.60_{\pm0.14}$ | $91.34_{\pm1.16}$ | $69.74_{\pm0.77}$ | $6.23_{\pm0.63}$ | $88.01_{\pm2.52}$ |
| | RoAST (Ours) | $97.13_{\pm0.09}$ | $92.31_{\pm0.24}$ | $74.25_{\pm0.11}$ | $4.46_{\pm0.03}$ | $89.74_{\pm0.25}$ |

Table 11: Robustness measurements of six different LMs fine-tuned on MNLI dataset for entailment task. All the values and error bars are mean and standard deviation across 3 random seeds.

| Type of LM | Method | $\text{Acc}_{\texttt{in}}$ | $\text{Acc}_{\texttt{shift}}$ | $\text{Acc}_{\texttt{adv}}$ | ECE ($\downarrow$) | AUROC |
|---|---|---|---|---|---|---|
| BERT | Vanilla | $86.25_{\pm0.11}$ | $60.20_{\pm0.59}$ | $39.25_{\pm0.30}$ | $23.03_{\pm0.58}$ | $70.88_{\pm1.64}$ |
| | AdvEmbed | $86.99_{\pm0.09}$ | $60.77_{\pm0.25}$ | $42.90_{\pm0.37}$ | $18.50_{\pm2.31}$ | $81.98_{\pm3.37}$ |
| | WConsol | $86.27_{\pm0.26}$ | $60.52_{\pm0.34}$ | $39.34_{\pm0.51}$ | $17.38_{\pm4.21}$ | $75.71_{\pm5.03}$ |
| | ROAST (Ours) | $86.89_{\pm0.06}$ | $60.38_{\pm0.61}$ | $42.58_{\pm0.47}$ | $14.97_{\pm0.38}$ | $82.41_{\pm0.80}$ |
| RoBERTa | Vanilla | $89.97_{\pm0.04}$ | $64.31_{\pm0.58}$ | $48.60_{\pm1.31}$ | $12.64_{\pm0.79}$ | $92.09_{\pm2.26}$ |
| | AdvEmbed | $90.24_{\pm0.07}$ | $64.23_{\pm0.38}$ | $50.20_{\pm0.72}$ | $12.48_{\pm0.71}$ | $93.83_{\pm0.52}$ |
| | WConsol | $90.54_{\pm0.01}$ | $65.04_{\pm0.55}$ | $49.00_{\pm0.10}$ | $10.84_{\pm0.90}$ | $91.49_{\pm1.40}$ |
| | ROAST (Ours) | $90.64_{\pm0.11}$ | $63.95_{\pm0.51}$ | $51.33_{\pm0.34}$ | $11.02_{\pm0.45}$ | $93.25_{\pm0.15}$ |
| ALBERT | Vanilla | $90.62_{\pm0.18}$ | $64.94_{\pm0.33}$ | $58.79_{\pm0.21}$ | $10.25_{\pm0.92}$ | $83.14_{\pm0.80}$ |
| | AdvEmbed | $90.60_{\pm0.15}$ | $64.19_{\pm0.14}$ | $58.82_{\pm0.25}$ | $10.92_{\pm0.68}$ | $86.63_{\pm4.49}$ |
| | WConsol | $90.43_{\pm0.20}$ | $64.37_{\pm0.35}$ | $56.14_{\pm1.95}$ | $9.67_{\pm0.08}$ | $80.65_{\pm3.98}$ |
| | ROAST (Ours) | $90.72_{\pm0.10}$ | $66.57_{\pm0.03}$ | $59.24_{\pm0.20}$ | $11.83_{\pm4.61}$ | $91.47_{\pm2.63}$ |
| XLNet | Vanilla | $89.29_{\pm0.10}$ | $63.74_{\pm0.23}$ | $49.33_{\pm0.36}$ | $14.04_{\pm1.66}$ | $77.53_{\pm8.00}$ |
| | AdvEmbed | $89.46_{\pm0.09}$ | $63.56_{\pm0.94}$ | $50.52_{\pm0.90}$ | $12.69_{\pm0.54}$ | $77.31_{\pm1.67}$ |
| | WConsol | $89.33_{\pm0.04}$ | $63.55_{\pm0.08}$ | $49.91_{\pm0.30}$ | $14.64_{\pm1.37}$ | $81.35_{\pm7.83}$ |
| | ROAST (Ours) | $90.03_{\pm0.04}$ | $63.69_{\pm0.52}$ | $50.11_{\pm0.14}$ | $10.28_{\pm0.37}$ | $78.53_{\pm3.07}$ |
| ELECTRA | Vanilla | $90.68_{\pm0.05}$ | $66.20_{\pm0.90}$ | $56.06_{\pm1.16}$ | $13.87_{\pm5.47}$ | $82.75_{\pm2.79}$ |
| | AdvEmbed | $90.64_{\pm0.01}$ | $65.66_{\pm0.57}$ | $56.95_{\pm0.91}$ | $12.96_{\pm2.51}$ | $88.43_{\pm0.11}$ |
| | WConsol | $90.51_{\pm0.13}$ | $67.21_{\pm0.41}$ | $55.43_{\pm0.84}$ | $15.36_{\pm5.44}$ | $84.05_{\pm4.02}$ |
| | ROAST (Ours) | $91.07_{\pm0.07}$ | $64.80_{\pm0.46}$ | $58.38_{\pm0.68}$ | $10.17_{\pm0.40}$ | $81.01_{\pm2.16}$ |
| DeBERTa | Vanilla | $90.09_{\pm0.15}$ | $65.01_{\pm0.80}$ | $52.61_{\pm1.39}$ | $15.87_{\pm7.81}$ | $87.89_{\pm2.40}$ |
| | AdvEmbed | $90.43_{\pm0.04}$ | $65.73_{\pm1.02}$ | $54.91_{\pm0.30}$ | $13.29_{\pm1.94}$ | $86.37_{\pm2.27}$ |
| | WConsol | $90.02_{\pm0.18}$ | $64.83_{\pm1.38}$ | $53.67_{\pm0.08}$ | $14.51_{\pm7.15}$ | $89.03_{\pm0.83}$ |
| | ROAST (Ours) | $90.97_{\pm0.09}$ | $66.07_{\pm0.31}$ | $56.53_{\pm0.66}$ | $9.94_{\pm0.34}$ | $88.15_{\pm1.73}$ |

Table 12: Robustness with different language models. Average relative improvements ($\Delta_{\texttt{avg}}$) compared to each vanilla fine-tuned LM are reported for each LM. All the values are mean across 3 random seeds.

| Model | Entailment | | | | Sentiment Classification | | | |
|---|---|---|---|---|---|---|---|---|
| | Vanilla | AdvEmbed | WCons | ROAST | Vanilla | AdvEmbed | WCons | ROAST |
| BERT-large | 0.00 | 22.85 | 20.23 | **25.34** | 0.00 | 18.24 | 15.85 | **22.76** |
| RoBERTa-large | 0.00 | 5.72 | 2.99 | **7.63** | 0.00 | 13.70 | 15.47 | **18.39** |
| ALBERT-xxlarge | 0.00 | 2.60 | -3.83 | **8.19** | 0.00 | 21.34 | 18.25 | **30.62** |
| XLNet-large | 0.00 | 2.44 | 2.75 | **3.30** | 0.00 | 1.44 | -1.76 | **12.74** |
| ELECTRA-large | 0.00 | **7.89** | -0.70 | 4.39 | 0.00 | 6.44 | -2.83 | **8.29** |
| DeBERTa-large | 0.00 | 2.81 | 3.79 | **11.93** | 0.00 | 11.95 | 8.25 | **18.90** |

Table 13: Robustness with GPT, LM with Transformer-decoder architecture. Average relative improvements ($\Delta_{\texttt{avg}}$) compared to each vanilla fine-tuned GPT. All the values are mean across 3 random seeds.

| Method | $\text{Acc}_{\texttt{in}}$ | $\text{Acc}_{\texttt{shift}}$ | $\text{Acc}_{\texttt{adv}}$ | ECE ($\downarrow$) | AUROC | $\Delta_{\texttt{avg}}$ |
|---|---|---|---|---|---|---|
| Vanilla | 94.99 | 84.33 | 62.38 | 8.01 | 95.30 | 0.00 |
| AdvEmbed | 94.92 | 84.19 | 63.58 | 7.90 | 95.81 | 2.60 |
| WCons | 94.88 | 80.72 | 61.80 | 9.06 | 93.57 | -15.37 |
| ROAST (Ours) | 95.03 | 81.91 | 64.96 | 8.52 | 97.16 | 5.07 |

Table 14: Accuracy of RoBERTa-large fine-tuned using SST-2 dataset for sentiment classifcation task. One in-distribution validation set and five distribution shifted datasets are evaluated. All the values with larger font are mean across 3 random seeds. The values with smaller font and plus-minus sign (±) are corresponding variance. Numbers in bracket means the number of samples.

| Method | SST-2 (872) | Distribution Shifted Datasets | | | | |
| | | Yelp (20K) | IMDB (25K) | c-IMDB (2440) | Poem (359) | Amazon (100K) |
|---|---|---|---|---|---|---|
| Vanilla | 96.29 | 96.43 | 88.82 | 91.79 | 88.02 | 93.91 |
| | ±0.14 | ±0.22 | ±0.40 | ±0.45 | ±0.60 | ±0.11 |
| WordDrop | 96.44 | 96.31 | 88.50 | 89.06 | 82.37 | 93.52 |
| | ±0.03 | ±0.08 | ±0.33 | ±0.70 | ±2.51 | ±0.14 |
| R-Drop | 96.44 | 96.18 | 88.78 | 91.80 | 88.21 | 93.78 |
| | ±0.19 | ±0.11 | ±0.49 | ±0.21 | ±1.25 | ±0.07 |
| HiddenCut | 96.67 | 96.17 | 88.89 | 91.39 | 85.79 | 93.47 |
| | ±0.34 | ±0.18 | ±0.17 | ±0.45 | ±0.82 | ±0.20 |
| AdvWeight | 96.41 | 96.34 | 88.33 | 91.78 | 89.88 | 93.25 |
| | ±0.29 | ±0.11 | ±0.14 | ±0.20 | ±0.73 | ±0.04 |
| AdvEmbed | 96.48 | 96.34 | 89.21 | 91.42 | 87.93 | 93.87 |
| | ±0.05 | ±0.14 | ±0.10 | ±0.34 | ±1.25 | ±0.05 |
| FreeLB | 96.33 | 96.34 | 89.20 | 91.27 | 88.95 | 93.96 |
| | ±0.34 | ±0.49 | ±0.23 | ±0.64 | ±2.28 | ±0.05 |
| SMART | 96.86 | 96.23 | 88.49 | 90.50 | 89.04 | 93.18 |
| | ±0.05 | ±0.04 | ±0.16 | ±0.16 | ±1.60 | ±0.07 |
| RIFT | 96.41 | 95.63 | 88.14 | 89.25 | 81.80 | 92.93 |
| | ±0.05 | ±0.03 | ±0.21 | ±0.89 | ±0.47 | ±0.04 |
| ChildPTune | 96.56 | 96.69 | 89.53 | 91.93 | 86.26 | 94.33 |
| | ±0.04 | ±0.15 | ±0.11 | ±0.36 | ±0.95 | ±0.02 |
| R3F | 96.56 | 96.24 | 88.83 | 90.97 | 89.23 | 93.67 |
| | ±0.09 | ±0.22 | ±0.16 | ±0.57 | ±0.66 | ±0.12 |
| WConsol | 96.60 | 97.02 | 89.75 | 91.93 | 87.56 | 94.49 |
| | ±0.22 | ±0.07 | ±0.30 | ±0.23 | ±1.93 | ±0.14 |
| LP-FT | 96.33 | 97.54 | 89.84 | 90.04 | 87.28 | 94.57 |
| | ±0.25 | ±0.03 | ±0.37 | ±0.53 | ±1.51 | ±0.13 |
| RoAST (Ours) | 96.87 | 96.90 | 89.52 | 92.10 | 89.04 | 94.32 |
| | ±0.20 | ±0.28 | ±0.07 | ±0.50 | ±0.57 | ±0.15 |

Table 15: Expected Calibration Error (ECE) of RoBERTa-large fine-tuned using SST-2 dataset for sentiment classifcation task. One in-distribution validation set and five distribution shifted datasets are evaluated. All the values with larger font are mean across 3 random seeds. The values with smaller font and plus-minus sign (±) are corresponding variance. Numbers in bracket means the number of samples. Lower ECE value indicates the better calibration.

| | | Distribution Shifted Datasets | | | | |
| Method | SST-2 (872) | Yelp (20K) | IMDB (25K) | c-IMDB (2440) | Poem (359) | Amazon (100K) |
|---|---|---|---|---|---|---|
| Vanilla | 3.80 | 5.30 | 3.70 | 5.13 | 5.87 | 5.03 |
| | ±0.37 | ±1.66 | ±1.77 | ±1.33 | ±0.79 | ±1.27 |
| WordDrop | 5.93 | 6.97 | 9.77 | 10.13 | 8.57 | 7.80 |
| | ±1.23 | ±1.58 | ±1.54 | ±1.76 | ±1.28 | ±3.32 |
| R-Drop | 4.90 | 5.27 | 3.80 | 6.17 | 5.53 | 5.30 |
| | ±1.45 | ±0.17 | ±1.56 | ±2.36 | ±0.99 | ±2.69 |
| HiddenCut | 3.10 | 4.00 | 6.70 | 6.50 | 7.23 | 6.37 |
| | ±0.62 | ±0.85 | ±1.31 | ±0.16 | ±0.57 | ±1.36 |
| AdvWeight | 5.30 | 7.20 | 9.60 | 7.87 | 7.27 | 6.30 |
| | ±0.59 | ±1.00 | ±0.99 | ±0.71 | ±0.09 | ±1.51 |
| AdvEmbed | 3.27 | 2.77 | 4.07 | 5.43 | 5.73 | 5.33 |
| | ±0.19 | ±0.09 | ±0.40 | ±0.62 | ±0.66 | ±0.09 |
| FreeLB | 3.63 | 5.47 | 6.17 | 5.97 | 4.37 | 4.50 |
| | ±1.43 | ±2.15 | ±2.88 | ±0.61 | ±0.59 | ±1.51 |
| SMART | 4.30 | 7.73 | 7.50 | 6.60 | 5.93 | 5.73 |
| | ±0.49 | ±0.56 | ±1.55 | ±0.45 | ±0.66 | ±0.33 |
| RIFT | 4.40 | 4.97 | 7.97 | 7.20 | 8.63 | 5.57 |
| | ±0.70 | ±2.58 | ±1.60 | ±0.36 | ±0.82 | ±1.11 |
| ChildPTune | 3.33 | 4.43 | 2.40 | 3.87 | 5.33 | 4.20 |
| | ±0.34 | ±0.76 | ±0.67 | ±1.16 | ±0.26 | ±1.22 |
| R3F | 4.60 | 4.30 | 6.90 | 6.60 | 5.70 | 6.13 |
| | ±0.86 | ±1.24 | ±1.70 | ±0.50 | ±0.86 | ±1.21 |
| WConsol | 2.50 | 3.00 | 4.20 | 5.00 | 6.13 | 5.57 |
| | ±0.24 | ±0.36 | ±0.73 | ±1.02 | ±1.48 | ±1.41 |
| LP-FT | 3.83 | 2.27 | 1.83 | 2.60 | 6.43 | 4.00 |
| | ±0.17 | ±0.12 | ±0.24 | ±0.24 | ±1.01 | ±0.22 |
| RoAST (Ours) | 3.50 | 4.03 | 7.17 | 7.07 | 6.90 | 4.80 |
| | ±0.49 | ±0.80 | ±1.32 | ±0.26 | ±1.61 | ±1.63 |

Table 16: Accuracy of RoBERTa-large fine-tuned using SST-2 dataset for sentiment classification task. Five adversarially constructed datasets are evaluated. All the values with larger font are mean across 3 random seeds. The values with smaller font and plus-minus sign (±) are corresponding variance. Numbers in bracket means the number of samples.

| Method | Adversarially Constructed Datasets | | | | |
| --- | --- | --- | --- | --- | --- |
| | TF-B (694) | TF-R (686) | Dynasent-R1 (4800) | Dynasent-R2 (960) | AdvGLUE (148) |
| Vanilla | 73.05 ±0.62 | 45.87 ±1.82 | 78.19 ±0.58 | 77.64 ±0.30 | 56.76 ±1.66 |
| WordDrop | 77.38 ±0.47 | 60.50 ±1.09 | 76.34 ±0.37 | 76.35 ±0.08 | 56.76 ±2.40 |
| R-Drop | 75.50 ±1.71 | 58.16 ±2.08 | 77.85 ±0.65 | 77.43 ±0.05 | 56.08 ±4.52 |
| HiddenCut | 78.15 ±0.95 | 60.88 ±1.07 | 76.33 ±0.57 | 77.01 ±0.26 | 59.23 ±1.77 |
| AdvWeight | 73.01 ±0.36 | 53.26 ±0.84 | 75.30 ±0.23 | 76.25 ±0.47 | 49.55 ±2.09 |
| AdvEmbed | 75.36 ±0.62 | 61.03 ±0.97 | 77.23 ±0.36 | 77.57 ±0.91 | 58.33 ±2.23 |
| FreeLB | 74.54 ±0.59 | 57.92 ±2.48 | 77.94 ±0.26 | 77.53 ±0.79 | 62.39 ±2.83 |
| SMART | 77.95 ±0.62 | 65.31 ±1.37 | 75.82 ±0.33 | 76.22 ±0.30 | 55.18 ±1.59 |
| RIFT | 77.14 ±0.41 | 67.35 ±2.75 | 74.35 ±0.62 | 77.33 ±1.01 | 57.21 ±1.77 |
| ChildPTune | 75.55 ±1.19 | 54.86 ±1.92 | 79.57 ±0.83 | 78.92 ±0.18 | 58.78 ±2.53 |
| R3F | 76.03 ±1.01 | 58.94 ±2.00 | 77.04 ±0.44 | 77.12 ±0.21 | 56.53 ±4.59 |
| WConsol | 76.70 ±0.58 | 55.64 ±0.66 | 80.75 ±0.31 | 79.96 ±0.50 | 61.26 ±2.49 |
| LP-FT | 76.32 ±0.77 | 57.82 ±0.72 | 81.92 ±0.37 | 81.04 ±0.44 | 65.31 ±0.64 |
| RoAST (Ours) | 77.09 ±0.24 | 61.18 ±0.86 | 79.79 ±0.41 | 79.27 ±0.47 | 65.54 ±2.40 |

Table 17: Expected Calibration Error (ECE) of RoBERTa-large fine-tuned using SST-2 dataset for sentiment classification task. Five adversarially constructed datasets are evaluated. All the values with larger font are mean across 3 random seeds. The values with smaller font and plus-minus sign ($\pm$) are corresponding variance. Numbers in bracket means the number of samples. Lower ECE value indicates the better calibration.

| Method | Adversarially Constructed Datasets | | | | |
| --- | --- | --- | --- | --- | --- |
| | TF-B (694) | TF-R (686) | Dynasent-R1 (4800) | Dynasent-R2 (960) | AdvGLUE (148) |
| Vanilla | 5.57 $\pm$1.93 | 19.73 $\pm$4.63 | 5.53 $\pm$1.41 | 4.90 $\pm$0.45 | 13.60 $\pm$5.06 |
| WordDrop | 8.53 $\pm$1.84 | 3.40 $\pm$0.14 | 6.97 $\pm$2.38 | 7.57 $\pm$1.50 | 4.97 $\pm$0.86 |
| R-Drop | 5.30 $\pm$1.59 | 8.43 $\pm$3.04 | 5.40 $\pm$2.81 | 5.80 $\pm$0.50 | 10.63 $\pm$3.60 |
| HiddenCut | 5.90 $\pm$1.13 | 4.43 $\pm$0.74 | 2.70 $\pm$0.96 | 2.87 $\pm$0.09 | 10.40 $\pm$1.66 |
| AdvWeight | 5.43 $\pm$0.52 | 8.30 $\pm$1.28 | 4.17 $\pm$0.62 | 3.90 $\pm$1.27 | 15.63 $\pm$4.15 |
| AdvEmbed | 3.97 $\pm$0.78 | 9.10 $\pm$1.77 | 3.23 $\pm$0.12 | 4.50 $\pm$1.07 | 13.17 $\pm$2.10 |
| FreeLB | 6.43 $\pm$0.17 | 13.70 $\pm$6.25 | 3.93 $\pm$2.07 | 5.47 $\pm$0.98 | 11.77 $\pm$5.00 |
| SMART | 7.53 $\pm$0.33 | 6.53 $\pm$0.78 | 3.70 $\pm$0.16 | 4.37 $\pm$0.90 | 9.57 $\pm$3.80 |
| RIFT | 11.43 $\pm$0.46 | 11.13 $\pm$2.41 | 2.73 $\pm$0.74 | 3.17 $\pm$0.24 | 8.10 $\pm$1.14 |
| ChildPTune | 4.13 $\pm$0.87 | 12.93 $\pm$0.57 | 2.77 $\pm$0.19 | 3.73 $\pm$0.70 | 14.17 $\pm$3.27 |
| R3F | 5.07 $\pm$1.60 | 6.77 $\pm$2.16 | 2.43 $\pm$0.61 | 3.53 $\pm$0.17 | 12.10 $\pm$2.14 |
| WConsol | 6.50 $\pm$1.00 | 8.10 $\pm$0.96 | 3.50 $\pm$0.36 | 3.27 $\pm$0.66 | 7.33 $\pm$0.53 |
| LP-FT | 5.13 $\pm$1.17 | 6.00 $\pm$0.41 | 3.33 $\pm$0.12 | 3.27 $\pm$0.34 | 5.80 $\pm$0.43 |
| RoAST (Ours) | 6.00 $\pm$0.94 | 5.13 $\pm$0.90 | 4.37 $\pm$0.81 | 4.70 $\pm$0.24 | 6.30 $\pm$0.59 |

Table 18: Anomaly detection performance (AUROC) of RoBERTa-large fine-tuned using SST-2 dataset for sentiment classification task. Six anomaly datasets are evaluated. All the values with larger font are mean across 3 random seeds. The values with smaller font and plus-minus sign (±) are corresponding variance. Numbers in bracket means the number of samples.

| Method | Anomaly Datasets | | | | | |
| | WMT16 (2999) | Multi30K (3071) | 20News (592) | QQP (40K) | MNLI-m (9815) | MNLI-mm (9832) |
|---|---|---|---|---|---|---|
| Vanilla | 83.47 | 84.87 | 96.03 | 87.33 | 84.17 | 84.47 |
| | ±4.41 | ±7.05 | ±0.45 | ±3.65 | ±3.86 | ±4.23 |
| WordDrop | 85.40 | 88.03 | 93.90 | 86.47 | 86.20 | 85.40 |
| | ±3.19 | ±2.66 | ±0.54 | ±1.39 | ±2.87 | ±2.86 |
| R-Drop | 86.07 | 90.87 | 96.33 | 89.97 | 86.40 | 85.53 |
| | ±1.03 | ±1.86 | ±0.17 | ±1.78 | ±1.21 | ±1,69 |
| HiddenCut | 87.90 | 92.53 | 95.40 | 89.23 | 87.47 | 86.90 |
| | ±0.08 | ±0.37 | ±0.99 | ±1.26 | ±0.25 | ±0.16 |
| AdvWeight | 84.60 | 90.77 | 92.93 | 88.67 | 85.07 | 84.77 |
| | ±1.31 | ±0.87 | ±0.29 | ±0.94 | ±1.30 | ±1.65 |
| AdvEmbed | 88.30 | 91.90 | 97.13 | 89.47 | 89.30 | 88.63 |
| | ±0.80 | ±1.18 | ±0.12 | ±1.38 | ±0.64 | ±0.82 |
| FreeLB | 86.97 | 89.63 | 96.37 | 91.03 | 87.43 | 87.50 |
| | ±0.17 | ±1.55 | ±2.09 | ±0.82 | ±0.44 | ±0.78 |
| SMART | 88.63 | 94.33 | 95.80 | 89.70 | 88.53 | 88.33 |
| | ±0.70 | ±0.31 | ±0.22 | ±0.22 | ±0.60 | ±0.66 |
| RIFT | 85.97 | 91.00 | 96.83 | 88.93 | 88.80 | 88.03 |
| | ±1.46 | ±1.14 | ±1.00 | ±1.53 | ±1.67 | ±2.33 |
| ChildPTune | 84.64 | 82.03 | 96.17 | 88.40 | 85.23 | 85.53 |
| | ±1.92 | ±3.84 | ±0.75 | ±1.41 | ±1.58 | ±1.76 |
| R3F | 84.90 | 91.33 | 95.57 | 87.80 | 86.13 | 85.23 |
| | ±0.79 | ±0.50 | ±0.74 | ±1.24 | ±0.45 | ±0.68 |
| WConsol | 87.50 | 90.27 | 96.30 | 90.30 | 86.90 | 86.37 |
| | ±1.31 | ±1.39 | ±0.37 | ±0.91 | ±1.18 | ±1.34 |
| LP-FT | 86.23 | 91.57 | 95.20 | 90.60 | 87.10 | 86.03 |
| | ±1.02 | ±0.73 | ±0.45 | ±0.86 | ±0.83 | ±1.23 |
| ROAST (Ours) | 88.60 | 91.30 | 95.43 | 90.87 | 88.17 | 87.83 |
| | ±0.54 | ±1.58 | ±0.47 | ±0.94 | ±0.65 | ±0.94 |

Table 19: Accuracy of RoBERTa-large fine-tuned using MNLI dataset for entailment task. Two in-distribution validation sets (MNLI-m and MNLI-mm) and seven distribution shifted datasets are evaluated. All the values with larger font are mean across 3 random seeds. The values with smaller font and plus-minus sign (±) are corresponding variance. Numbers in bracket means the number of samples.

| | | | Distribution Shifted Datasets | | | | | | |
|---|---|---|---|---|---|---|---|---|---|
| Method | MNLI-m (9815) | MNLI-mm (9832) | Diag (1104) | HANS* (30K) | QNLI* (5266) | WNLI* (635) | NQ-NLI* (4855) | FEVER-NLI (20K) | WANLI (5000) |
| Vanilla | 90.15 | 89.80 | 66.09 | 75.70 | 60.35 | 53.91 | 61.94 | 69.62 | 62.55 |
| | ±0.06 | ±0.12 | ±0.55 | ±0.71 | ±2.76 | ±1.45 | ±0.74 | ±0.40 | ±0.77 |
| WordDrop | 90.44 | 90.26 | 64.98 | 76.70 | 54.84 | 52.86 | 61.28 | 68.82 | 62.93 |
| | ±0.15 | ±0.24 | ±0.64 | ±1.63 | ±0.53 | ±0.20 | ±0.31 | ±0.44 | ±0.79 |
| R-Drop | 90.61 | 90.67 | 65.46 | 74.82 | 54.88 | 51.34 | 60.56 | 68.92 | 63.20 |
| | ±0.19 | ±0.23 | ±0.56 | ±0.69 | ±0.92 | ±0.72 | ±0.05 | ±0.35 | ±0.19 |
| HiddenCut | 90.62 | 90.35 | 66.76 | 76.75 | 54.55 | 53.75 | 62.11 | 69.14 | 63.79 |
| | ±0.23 | ±0.18 | ±0.39 | ±0.70 | ±0.69 | ±0.91 | ±0.23 | ±0.23 | ±0.32 |
| AdvWeight | 90.39 | 90.00 | 65.94 | 74.83 | 54.91 | 51.71 | 61.29 | 69.19 | 62.21 |
| | ±0.13 | ±0.15 | ±0.97 | ±1.56 | ±1.20 | ±0.83 | ±1.02 | ±0.39 | ±0.71 |
| AdvEmbed | 90.51 | 89.97 | 67.75 | 77.77 | 55.77 | 53.23 | 61.81 | 69.29 | 63.99 |
| | ±0.03 | ±0.12 | ±0.94 | ±0.56 | ±2.64 | ±0.45 | ±0.26 | ±0.03 | ±0.60 |
| FreeLB | 90.38 | 90.16 | 67.39 | 79.47 | 55.22 | 56.30 | 62.62 | 69.44 | 64.12 |
| | ±0.08 | ±0.13 | ±0.09 | ±0.24 | ±0.65 | ±0.08 | ±0.08 | ±0.12 | ±0.14 |
| SMART | 90.78 | 90.69 | 65.53 | 77.73 | 53.95 | 51.42 | 60.87 | 69.82 | 63.57 |
| | ±0.05 | ±0.04 | ±0.23 | ±0.53 | ±0.03 | ±0.24 | ±0.12 | ±0.11 | ±0.11 |
| RIFT | 89.75 | 89.69 | 65.90 | 67.22 | 57.85 | 52.17 | 60.79 | 69.01 | 63.00 |
| | ±0.25 | ±0.13 | ±0.51 | ±0.83 | ±1.80 | ±1.13 | ±0.84 | ±0.42 | ±0.29 |
| ChildPTune | 90.14 | 90.02 | 65.94 | 75.19 | 57.83 | 53.86 | 62.38 | 69.46 | 63.96 |
| | ±0.06 | ±0.07 | ±0.63 | ±2.45 | ±1.59 | ±1.29 | ±0.88 | ±0.47 | ±0.58 |
| R3F | 90.57 | 90.25 | 65.55 | 76.75 | 56.54 | 52.55 | 61.53 | 69.10 | 62.75 |
| | ±0.12 | ±0.02 | ±1.04 | ±1.90 | ±4.32 | ±0.30 | ±0.78 | ±0.24 | ±0.80 |
| WConsol | 90.71 | 90.37 | 67.09 | 76.52 | 61.64 | 55.91 | 61.28 | 70.15 | 62.72 |
| | ±0.08 | ±0.07 | ±0.50 | ±1.65 | ±2.43 | ±0.68 | ±0.53 | ±0.25 | ±0.42 |
| LP-FT | 90.57 | 90.28 | 67.60 | 77.12 | 56.85 | 55.59 | 62.28 | 69.86 | 63.63 |
| | ±0.15 | ±0.13 | ±0.30 | ±1.40 | ±1.37 | ±1.70 | ±0.97 | ±0.15 | ±0.58 |
| RoAST (Ours) | 90.78 | 90.50 | 67.18 | 75.82 | 58.26 | 53.75 | 60.24 | 69.03 | 63.34 |
| | ±0.08 | ±0.21 | ±0.86 | ±2.06 | ±3.44 | ±0.45 | ±0.98 | ±0.61 | ±0.88 |

Table 20: Expected Calibration Error (ECE) of RoBERTa-large fine-tuned using MNLI dataset for entailment task. Two in-distribution validation sets (MNLI-m and MNLI-mm) and seven distribution shifted datasets are evaluated. All the values with larger font are mean across 3 random seeds. The values with smaller font and plus-minus sign (±) are corresponding variance. Numbers in bracket means the number of samples. Lower ECE value indicates the better calibration.

| | | | Distribution Shifted Datasets | | | | | | |
|---|---|---|---|---|---|---|---|---|---|
| Method | MNLI-m (9815) | MNLI-mm (9832) | Diag (1104) | HANS* (30K) | QNLI* (5266) | WNLI* (635) | NQ-NLI* (4855) | FEVER-NLI (20K) | WANLI (5000) |
| Vanilla | 10.10 | 11.03 | 4.73 | 8.67 | 27.07 | 9.17 | 34.23 | 4.70 | 3.03 |
| | ±0.96 | ±2.29 | ±0.90 | ±1.40 | ±2.44 | ±2.21 | ±2.45 | ±0.62 | ±1.80 |
| WordDrop | 12.80 | 12.33 | 6.97 | 13.17 | 24.43 | 3.47 | 24.70 | 8.73 | 5.80 |
| | ±0.94 | ±1.11 | ±0.77 | ±1.18 | ±0.76 | ±0.31 | ±0.51 | ±1.11 | ±0.36 |
| R-Drop | 8.73 | 8.87 | 7.87 | 6.97 | 24.83 | 7.40 | 26.30 | 5.13 | 4.97 |
| | ±0.25 | ±0.45 | ±1.19 | ±0.74 | ±2.89 | ±0.51 | ±0.65 | ±0.41 | ±0.12 |
| HiddenCut | 11.70 | 12.83 | 6.70 | 9.17 | 26.40 | 7.13 | 30.20 | 5.67 | 1.70 |
| | ±1.28 | ±0.33 | ±1.12 | ±0.68 | ±0.92 | ±0.77 | ±1.02 | ±0.29 | ±0.08 |
| AdvWeight | 11.40 | 11.33 | 4.83 | 8.97 | 28.03 | 7.87 | 29.73 | 6.13 | 4.40 |
| | ±1.92 | ±1.36 | ±1.54 | ±1.03 | ±3.64 | ±1.28 | ±3.27 | ±1.10 | ±1.10 |
| AdvEmbed | 8.87 | 9.23 | 4.37 | 9.13 | 31.00 | 10.33 | 34.13 | 3.37 | 4.30 |
| | ±2.00 | ±2.61 | ±0.60 | ±0.31 | ±4.41 | ±3.23 | ±4.71 | ±1.52 | ±1.47 |
| FreeLB | 9.60 | 8.70 | 4.40 | 10.35 | 27.80 | 7.75 | 32.75 | 5.45 | 3.45 |
| | ±1.50 | ±0.50 | ±1.00 | ±4.65 | ±7.40 | ±1.45 | ±3.95 | ±0.65 | ±0.25 |
| SMART | 10.30 | 10.20 | 7.05 | 5.05 | 25.55 | 7.00 | 26.65 | 5.00 | 6.30 |
| | ±0.23 | ±0.18 | ±0.05 | ±0.25 | ±0.35 | ±3.23 | ±0.15 | ±1.23 | ±0.53 |
| RIFT | 12.50 | 12.48 | 4.20 | 12.20 | 29.83 | 7.48 | 30.93 | 5.53 | 2.00 |
| | ±0.86 | ±0.99 | ±0.25 | ±1.59 | ±3.11 | ±1.72 | ±2.39 | ±0.60 | ±1.04 |
| ChildPTune | 8.03 | 7.87 | 4.83 | 8.70 | 29.40 | 10.37 | 35.13 | 5.67 | 4.30 |
| | ±3.96 | ±3.88 | ±0.92 | ±2.62 | ±8.36 | ±3.62 | ±6.26 | ±0.62 | ±2.07 |
| R3F | 9.93 | 10.73 | 4.90 | 8.43 | 23.33 | 7.80 | 29.00 | 5.03 | 3.33 |
| | ±1.62 | ±1.48 | ±1.75 | ±1.33 | ±10.62 | ±1.18 | ±4.00 | ±1.35 | ±0.93 |
| WConsol | 8.77 | 8.57 | 5.93 | 7.23 | 19.90 | 5.87 | 28.27 | 5.00 | 3.20 |
| | ±1.64 | ±1.70 | ±0.48 | ±2.19 | ±3.15 | ±0.25 | ±2.32 | ±1.44 | ±1.28 |
| LP-FT | 11.03 | 10.87 | 5.40 | 11.13 | 26.57 | 6.90 | 32.67 | 4.03 | 2.57 |
| | ±0.70 | ±0.71 | ±1.63 | ±0.77 | ±3.35 | ±0.83 | ±1.62 | ±1.20 | ±0.61 |
| ROAST (Ours) | 7.40 | 7.43 | 6.47 | 10.30 | 22.00 | 7.27 | 26.83 | 5.73 | 3.97 |
| | ±0.29 | ±0.56 | ±0.49 | ±2.13 | ±6.84 | ±0.74 | ±3.38 | ±1.70 | ±0.60 |

Table 21: Accuracy of RoBERTa-large fine-tuned using MNLI dataset for entailment task. Nine adversarially constructed datasets are evaluated. All the values with larger font are mean across 3 random seeds. The values with smaller font and plus-minus sign (±) are corresponding variance. Numbers in bracket means the number of samples.

| Method | Adversarially Constructed Datasets | | | | | | | | |
|---|---|---|---|---|---|---|---|---|---|
| | TF-B-m (772) | TF-B-mm (746) | TF-R-m (775) | TF-R-mm (775) | ANLI-R1 (1000) | ANLI-R2 (1000) | ANLI-R3 (1200) | AdvGLUE-m (121) | AdvGLUE-mm (162) |
| Vanilla | 71.11 | 68.36 | 50.84 | 52.52 | 42 | 28.70 | 27.56 | 55.37 | 40.95 |
| | ±0.42 | ±1.26 | ±2.76 | ±1.90 | ±2.79 | ±0.51 | ±0.51 | ±2.34 | ±2.38 |
| WordDrop | 74.61 | 73.28 | 58.06 | 61.63 | 42.77 | 30.33 | 26.61 | 62.81 | 41.98 |
| | ±1.04 | ±0.17 | ±0.32 | ±0.16 | ±0.79 | ±1.27 | ±0.86 | ±1.79 | ±3.07 |
| R-Drop | 73.36 | 74.04 | 59.57 | 63.48 | 42.47 | 29.37 | 27.39 | 56.75 | 34.77 |
| | ±0.24 | ±1.35 | ±0.80 | ±1.70 | ±1.51 | ±0.98 | ±0.34 | ±2.55 | ±1.05 |
| HiddenCut | 73.06 | 70.46 | 55.74 | 57.98 | 43.07 | 29.67 | 26.86 | 58.13 | 38.89 |
| | ±1.50 | ±0.71 | ±0.18 | ±1.72 | ±0.86 | ±0.79 | ±0.75 | ±1.95 | ±0.87 |
| AdvWeight | 70.03 | 68.72 | 52.47 | 53.59 | 42.30 | 27.60 | 27.78 | 52.62 | 36.83 |
| | ±1.66 | ±1.66 | ±2.48 | ±2.26 | ±0.71 | ±0.29 | ±1.19 | ±0.39 | ±1.54 |
| AdvEmbed | 71.76 | 71.13 | 54.02 | 57.03 | 44.43 | 29.60 | 28.14 | 59.23 | 36.42 |
| | ±0.56 | ±0.46 | ±2.25 | ±0.46 | ±0.61 | ±0.50 | ±0.39 | ±2.81 | ±4.31 |
| FreeLB | 70.27 | 69.37 | 52.06 | 50.19 | 45.45 | 27.70 | 28.00 | 61.57 | 40.74 |
| | ±1.10 | ±1.27 | ±2.77 | ±3.10 | ±3.15 | ±0.40 | ±0.83 | ±0.41 | ±1.23 |
| SMART | 73.32 | 74.80 | 58.97 | 64.71 | 43.85 | 31.20 | 28.46 | 57.44 | 37.65 |
| | ±0.13 | ±0.67 | ±1.42 | ±0.45 | ±0.75 | ±0.30 | ±0.13 | ±0.41 | ±1.23 |
| RIFT | 77.08 | 75.74 | 65.13 | 66.13 | 43.45 | 28.15 | 26.83 | 59.50 | 39.97 |
| | ±2.11 | ±1.68 | ±4.10 | ±3.59 | ±0.51 | ±1.06 | ±0.19 | ±2.26 | ±1.41 |
| ChildPTune | 67.75 | 66.67 | 47.01 | 48.17 | 44.10 | 25.87 | 26.22 | 54.27 | 38.27 |
| | ±1.70 | ±1.97 | ±2.41 | ±4.30 | ±1.02 | ±0.98 | ±0.98 | ±1.56 | ±0.50 |
| R3F | 72.93 | 71.49 | 56.64 | 57.98 | 41.80 | 27.97 | 26.67 | 56.20 | 40.95 |
| | ±0.92 | ±0.17 | ±1.95 | ±1.23 | ±1.66 | ±0.54 | ±0.65 | ±2.94 | ±0.58 |
| WConsol | 72.06 | 71.36 | 51.87 | 50.97 | 46.90 | 26.20 | 24.86 | 56.47 | 40.33 |
| | ±1.25 | ±0.96 | ±1.66 | ±1.53 | ±1.51 | ±0.42 | ±0.67 | ±1.03 | ±2.27 |
| LP-FT | 70.90 | 70.24 | 49.76 | 51.23 | 45.90 | 27.07 | 23.69 | 57.58 | 43.00 |
| | ±1.33 | ±1.52 | ±1.78 | ±4.10 | ±1.49 | ±0.21 | ±0.79 | ±1.40 | ±0.77 |
| ROAST (Ours) | 75.60 | 72.74 | 56.30 | 56.60 | 45.23 | 28.00 | 25.92 | 58.95 | 42.59 |
| | ±1.44 | ±0.84 | ±1.33 | ±2.26 | ±1.93 | ±0.14 | ±0.42 | ±0.78 | ±2.02 |

Table 22: Expected Calibration Error (ECE) of RoBERTa-large fine-tuned using MNLI dataset for entailment task. Nine adversarially constructed datasets are evaluated. All the values with larger font are mean across 3 random seeds. The values with smaller font and plus-minus sign (±) are corresponding variance. Numbers in bracket means the number of samples. Lower ECE value indicates the better calibration.

| | Adversarially Constructed Datasets | | | | | | | | |
| Method | TF-B-m (772) | TF-B-mm (746) | TF-R-m (775) | TF-R-mm (775) | ANLI-R1 (1000) | ANLI-R2 (1000) | ANLI-R3 (1200) | AdvGLUE-m (121) | AdvGLUE-mm (162) |
|---|---|---|---|---|---|---|---|---|---|
| Vanilla | 6.80 | 6.93 | 6.80 | 5.43 | 12.80 | 25.93 | 26.17 | 7.97 | 16.00 |
| | ±1.06 | ±1.58 | ±0.00 | ±0.17 | ±4.70 | ±2.31 | ±2.99 | ±2.17 | ±1.88 |
| WordDrop | 13.43 | 12.83 | 7.57 | 8.17 | 6.13 | 17.47 | 20.77 | 12.57 | 7.70 |
| | ±0.70 | ±0.56 | ±0.39 | ±0.88 | ±0.54 | ±1.55 | ±1.46 | ±1.50 | ±1.02 |
| R-Drop | 11.40 | 12.50 | 6.73 | 7.67 | 8.90 | 19.40 | 20.37 | 9.73 | 13.30 |
| | ±0.88 | ±0.86 | ±1.18 | ±1.70 | ±1.16 | ±1.27 | ±0.24 | ±2.81 | ±0.90 |
| HiddenCut | 7.93 | 6.23 | 3.17 | 4.83 | 10.53 | 23.17 | 24.90 | 8.20 | 12.43 |
| | ±0.76 | ±1.30 | ±0.60 | ±0.21 | ±1.04 | ±0.82 | ±0.59 | ±0.86 | ±0.71 |
| AdvWeight | 8.97 | 9.30 | 6.80 | 5.67 | 10.63 | 23.03 | 22.03 | 8.83 | 14.90 |
| | ±3.43 | ±2.67 | ±1.34 | ±1.48 | ±2.43 | ±3.51 | ±2.45 | ±0.62 | ±3.28 |
| AdvEmbed | 5.83 | 6.03 | 4.87 | 5.20 | 11.00 | 25.50 | 26.47 | 7.37 | 18.40 |
| | ±1.48 | ±2.32 | ±1.56 | ±1.28 | ±3.63 | ±3.92 | ±4.03 | ±1.22 | ±1.28 |
| FreeLB | 6.70 | 6.05 | 7.10 | 6.55 | 8.15 | 26.30 | 26.65 | 9.55 | 13.30 |
| | ±1.80 | ±1.85 | ±1.20 | ±3.35 | ±0.95 | ±2.80 | ±2.85 | ±2.95 | ±1.80 |
| SMART | 12.15 | 13.35 | 8.25 | 10.40 | 5.35 | 16.50 | 17.55 | 10.35 | 8.45 |
| | ±0.55 | ±1.05 | ±0.85 | ±0.70 | ±0.15 | ±0.50 | ±0.15 | ±0.85 | ±1.15 |
| RIFT | 11.40 | 9.95 | 8.68 | 8.05 | 11.15 | 26.68 | 26.43 | 9.53 | 14.13 |
| | ±1.86 | ±2.31 | ±1.38 | ±0.87 | ±0.78 | ±1.37 | ±0.59 | ±1.56 | ±0.95 |
| ChildPTune | 7.20 | 6.97 | 10.13 | 11.10 | 11.43 | 28.90 | 28.00 | 9.73 | 17.57 |
| | ±1.87 | ±1.93 | ±6.32 | ±6.37 | ±7.38 | ±8.81 | ±8.67 | ±1.03 | ±7.33 |
| R3F | 8.27 | 8.20 | 4.77 | 5.27 | 9.47 | 22.80 | 23.47 | 10.03 | 10.17 |
| | ±0.92 | ±2.30 | ±1.35 | ±1.13 | ±2.27 | ±2.91 | ±4.00 | ±1.01 | ±3.30 |
| WConsol | 10.63 | 11.10 | 6.27 | 4.83 | 5.43 | 22.27 | 22.73 | 10.63 | 8.43 |
| | ±2.255 | ±1.43 | ±0.65 | ±0.93 | ±1.77 | ±3.21 | ±2.26 | ±1.31 | ±0.66 |
| LP-FT | 7.17 | 6.53 | 6.57 | 6.60 | 9.00 | 27.30 | 29.43 | 11.10 | 12.63 |
| | ±1.66 | ±0.45 | ±0.61 | ±1.31 | ±1.23 | ±2.15 | ±1.76 | ±0.54 | ±2.17 |
| ROAST (Ours) | 11.47 | 9.87 | 6.53 | 7.17 | 6.10 | 20.43 | 21.63 | 10.23 | 7.47 |
| | ±1.26 | ±1.22 | ±1.13 | ±1.35 | ±0.28 | ±1.60 | ±1.84 | ±1.92 | ±0.62 |

Table 23: Anomaly detection performance (AUROC) of RoBERTa-large fine-tuned using MNLI dataset for entailment task. Five anomaly datasets are evaluated. All the values with larger font are mean across 3 random seeds. The values with smaller font and plus-minus sign (±) are corresponding variance. Numbers in bracket means the number of samples.

| Method | Anomaly Datasets | | | | |
| --- | --- | --- | --- | --- | --- |
| | WMT16 (2999) | Multi30K (3071) | SST-2 (872) | 20News (592) | QQP (40K) |
| Vanilla | 94.93 | 98.23 | 97.50 | 85.67 | 84.10 |
| | ±2.65 | ±0.87 | ±1.06 | ±3.41 | ±4.11 |
| WordDrop | 92.00 | 99.47 | 97.97 | 74.97 | 75.43 |
| | ±1.21 | ±0.21 | ±0.62 | ±4.39 | ±0.88 |
| R-Drop | 96.03 | 98.50 | 98.30 | 84.67 | 81.97 |
| | ±1.06 | ±1.26 | ±0.57 | ±1.55 | ±1.59 |
| HiddenCut | 95.57 | 97.87 | 97.50 | 85.23 | 82.03 |
| | ±1.28 | ±1.68 | ±1.04 | ±2.17 | ±1.17 |
| AdvWeight | 96.70 | 99.43 | 98.70 | 78.37 | 81.87 |
| | ±0.92 | ±0.38 | ±0.45 | ±5.00 | ±0.82 |
| AdvEmbed | 96.93 | 99.00 | 98.23 | 88.60 | 86.40 |
| | ±0.53 | ±0.49 | ±0.09 | ±2.50 | ±1.04 |
| FreeLB | 98.15 | 99.35 | 98.50 | 84.45 | 85.75 |
| | ±0.05 | ±0.25 | ±0.10 | ±2.35 | ±2.35 |
| SMART | 94.55 | 99.30 | 96.10 | 87.20 | 79.95 |
| | ±0.95 | ±0.10 | ±0.50 | ±1.30 | ±0.85 |
| RIFT | 93.18 | 98.35 | 97.28 | 91.80 | 82.00 |
| | ±3.08 | ±1.35 | ±0.97 | ±1.52 | ±1.40 |
| ChildPTune | 82.87 | 95.90 | 87.43 | 83.57 | 83.27 |
| | ±14.91 | ±3.41 | ±7.23 | ±1.51 | ±3.81 |
| R3F | 93.63 | 99.07 | 95.73 | 87.20 | 83.40 |
| | ±2.10 | ±0.38 | ±2.88 | ±1.37 | ±2.38 |
| WConsol | 90.67 | 98.80 | 94.40 | 90.17 | 84.43 |
| | ±3.56 | ±0.29 | ±1.36 | ±0.47 | ±1.84 |
| LP-FT | 93.33 | 98.63 | 94.40 | 91.03 | 90.73 |
| | ±1.68 | ±0.40 | ±0.08 | ±1.65 | ±2.08 |
| RoAST (Ours) | 94.90 | 98.37 | 96.67 | 90.83 | 85.47 |
| | ±1.14 | ±0.61 | ±1.17 | ±1.88 | ±0.62 |