# OpenReview forum: "RoAST: Robustifying Language Models via Adversarial Perturbation with Selective Training"
_EMNLP/2023/Conference — EMNLP 2023 Findings_

### Official Review · Reviewer_Gq2X · 2023-08-02

**Soundness:** 4

**Excitement:**

3: Ambivalent: It has merits (e.g., it reports state-of-the-art results, the idea is nice), but there are key weaknesses (e.g., it describes incremental work), and it can significantly benefit from another round of revision. However, I won't object to accepting it if my co-reviewers champion it.

**Paper Topic And Main Contributions:**

The paper proposes an adaptive gradient update approach by reducing the update frequency of large variance parameters. The selective update strategy looks like "gradient dropout" by masking a gradient with a probability value p and rescaling it by 1/p if it is updated. Experimental results show that this effectively increases the robustness of the model.

**Questions For The Authors:**

- Can you show some results of FreeLB++, which runs adversarial perturbation for 10 steps instead of one? Li et al, EMNLP 2021, Searching for an effective defender: Benchmarking defense against adversarial word substitution.


**Reasons To Accept:**

- The experiments are comprehensive and the results are great.
- The approach has the potential to be a stronger and more generalised technique, rather than just a few NLP tasks.

**Reasons To Reject:**

- Further analysis is required to analyze why the importance score may benefit the model, as it is close to the update formula in Adam optimizer. In Table 4, the s(theta) row has better adversarial robustness, even better than the AT baseline. Why? The approach to computing s(theta) is in spirit to the second order momentum in Adam, which measures the variance of gradients. For parameters with large fluctuations, Adam will update with smaller steps. My question is, why does adversarial training with Adam perform worse than s(theta)? A more detailed study may make this clear. In addition, some explanations are not convincing enough and maybe more experiments would make them stronger.

- Why does using p(theta) even hurt performance when compared to s(theta) only?

- How does the integration of m(theta) increase so much? If this is because of preserving the parameters, does it indicate that training a model for 100 epochs will be less robust than 10 epochs only? Does it indicate that a sparse update paradigm without p(theta) as the denomination will also achieve some robustness?

- Some experimental results seem to be contrary to motivation. In Table 5, a random/min s(theta) still outperforms the baseline, especially for sentiment classification tasks. Does this mean that the importance score is not that important?

**Reproducibility:**

4: Could mostly reproduce the results, but there may be some variation because of sample variance or minor variations in their interpretation of the protocol or method.

**Reviewer Confidence:**

4: Quite sure. I tried to check the important points carefully. It's unlikely, though conceivable, that I missed something that should affect my ratings.

---

> ### Author Rebuttal · Authors · 2023-08-29
>
> We sincerely appreciate your efforts and insightful comments to improve the manuscript. We respond to each of your comments one-by-one in what follows. In tables, all the values are mean across 3 random seeds.
>
> ---
>
> #### **[R1]. Why does adversarial training with Adam perform worse than $s(\theta)$? A more detailed study may make this clear.**
> **[A].** Thank you for your insightful question. We'd like to provide some clarification here. As you pointed out, the introduced importance score $s(\theta)$ is closely related to the second-order momentum in Adam, as both utilize the square of gradients of parameters.
>
> However, the ways in which this gradient information is used are totally different; $s(\theta)$ in Table 4 (denoted in (c)) uses it to select the parameters to update via hard thresholding, i.e., the model is updated with sparsed gradients. On the other hand, Adam uses it to re-scale the gradients to update the parameters showing large fluctuations with smaller steps. In the context of incorporating the gradient information as the re-scaling term, $p(\theta)$ in Table 4 (denoted in (d)) is more closely related to Adam, as it also re-scales the gradients to update the model using $p(\theta)$, which is derived from $s(\theta)$ in Equation 3 (described in lines 504-511).
>
> As shown in Table 4, the incorporation of the proposed importance score via re-scaling, denoted by (d), performs slightly worse than adversarial training with Adam, denoted by (a). However, the incorporation of the importance score via a selective update, denoted by (c), outperforms both. This result indicates that the better robustness of $s(\theta)$ in Table 4 is not solely due to incorporating the importance score into the model update, but rather to the careful design of using them via selective training with sparse updates.
>
> To avoid potential confusion, we will clarify the naming of columns in Table 4 $\big(s(\theta), p(\theta), m(\theta)\big)$ to be more explicit in the revised draft. We will also incorporate the above discussion to make it clearer.
>
> ---
>
> #### **[R2]. Why does using $p(\theta)$ even hurt performance when compared to $s(\theta)$ only?**
> **[A].** First, we clarify that there is a key difference between the two approaches in Table 4, (c) and (d), regarding how they incorporate the importance score into the model update. (c) (i.e., $s(\theta)$ only) uses thresholding (i.e., sparsity) by only keeping the gradient of a relatively important parameter, which has a high value of $s(\theta)$, via hard thresholding. In contrast, (d) (i.e., using $p(\theta)$ from $s(\theta)$) updates the model by re-scaling gradients according to their relative importance in terms of the values of $p(\theta)$.
>
> As a result, the two approaches exhibit significantly different behavior. $s(\theta)$ can strictly preserve the original parameters of the pre-trained LM, which is less important to the given task, while $p(\theta)$ updates all the model parameters; therefore, the observed performance decrease with $p(\theta)$ compared to $s(\theta)$ could be due to this difference. Consequently, this result indicates the advantages of imposing sparsity rather than re-scaling to update the model for a given task while preserving the generalizable knowledge of the pre-trained LM.
>
> ---
>
> #### **[R3]. How does the integration of $m(\theta)$ increase so much? If this is because of preserving the parameters, does it indicate that training a model for 100 epochs will be less robust than 10 epochs only? Does it indicate that a sparse update paradigm without $p(\theta)$ as the denomination will also achieve some robustness?**
> **[A].** Thank you for the insightful questions and suggestions!
> - ***How does the integration of $m(\theta)$ increase so much?***: We first clarify that integrating $m(\theta)$ has two distinct advantages for training the model. Firstly, it preserves the parameters by selectively training them using masked (i.e., sparse) gradients. Secondly, it reduces distortion from the original gradients by sampling the mask instead of deterministic selection.
> Here, the gain from the first advantage is not only obtainable from $m(\theta)$. Hard thresholding via $s(\theta)$ also shares the same advantages of selective training and it can be verified with the results in Table 4, i.e., (c) > (a), (b), (d). However, it is important to note that its effectiveness can be limited since thresholding can distort the gradients from the original direction through deterministic updates of important parameters.
> Therefore, the second advantage of $m(\theta)$ is crucial, as it improves selective training by reducing the risk of distortion. By sampling the mask for updates from the distribution $p(\theta)$, which prioritizes important parameters, $m(\theta)$ continuously benefits from selective training while also covering overlooked parameters through stochastic sampling. This essential advantage of integrating $m(\theta)$ is demonstrated by improvements over (c,d) in Table 4.
> - ***Model for 100 epochs will be less robust than 10 epochs only?***: As per your suggestion, we performed additional experiments by training RoBERTa-large on SST-2, using the $\text{Vanilla}$ method with a constant learning rate for 100 epochs. As shown in the table below, the model's robustness significantly decreases as the training progresses. This outcome confirms that preserving the parameters is critical for maintaining model robustness, which is a fundamental principle of selective training with $m(\theta)$ in $\text{RoAST}$.
> | Method  | Acc$_{\tt in}$ | Acc$_{\tt shift}$ | Acc$_{\tt adv}$ | ECE ($\downarrow$) | AUROC | $\Delta_{\tt avg}$ |
> |:--------|:------------:|:----:|:------------:|:------------:|:----:|:------------:|
> | Vanilla (Orig)  | $96.29$ | $91.79$ | $66.30$ | $7.11$ | $86.72$ | $0.00$ |
> | Epoch 20  | $94.88$ | $90.66$ | $64.82$ | $7.75$ | $83.83$ | $-17.45$ |
> | Epoch 40  | $93.92$ | $88.52$ | $60.54$ | $9.81$ | $77.04$ | $-46.38$ |
> | Epoch 60 | $93.42$ | $88.11$ | $59.63$ | $10.16$ | $72.28$ | $-58.74$ |
> | Epoch 80 | $93.39$ | $87.43$ | $59.03$ | $10.68$ | $73.83$ | $-60.08$ |
> | Epoch 100 | $93.31$ | $87.52$ | $58.28$ | $10.62$ | $67.61$ | $-69.94$ |
>
> - ***A sparse update paradigm without $p(\theta)$ as the denomination will also achieve some robustness?***: To answer your question, we additionally train RoBERTa-large on SST-2, using $\text{RoAST}$ without the denomination term by $p(\theta)$ (in Equation 4). The results are presented in the table below.
> | Method  | Acc$_{\tt in}$ | Acc$_{\tt shift}$ | Acc$_{\tt adv}$ | ECE ($\downarrow$) | AUROC | $\Delta_{\tt avg}$ |
> |:--------|:------------:|:----:|:------------:|:------------:|:----:|:------------:|
> | $\text{Vanilla}$ | $96.29$ | $91.79$ | $66.30$ | $7.11$ | $86.72$ | $0.00$ |
> | $\text{RoAST}$ without $p(\theta)$ | $\underline{96.62}$ | $\underline{92.08}$ | $\underline{72.21}$ | $\bf 5.27$ | $\bf 90.99$ | $17.57$ |
> | $\text{RoAST}$ | $\bf 96.87$ | $\bf 92.38$ | $\bf 72.57$ | $\underline{5.45}$ | $\underline{90.37}$ | $\bf 18.39$ |
>
>   From the table, we can observe that a sparse update paradigm from $m(\theta)$ without $p(\theta)$ also achieves the strong multi-perspective robustness; this result implies that the aforementioned two advantages by introducing $m(\theta)$ is key of the improvement from RoAST, and the scaling via denomination provides additional gains upon that.
>
> We believe that these questions and responses provide many insights to understand our work. We will incorporate these in the revised manuscript accordingly.
>
> ---
>
> #### **[R4]. In Table 5, a random/min s(theta) still outperforms the baseline, especially for sentiment classification tasks. Does this mean that the importance score is not that important?**
> **[A].** Thank you for the question. Firstly, it is important to note that both the $\text{Rand}$ and $\text{Min}$ methods in Table 5 use not only selective training with varying masking strategies, but also adversarial perturbation during training. Therefore, a more appropriate comparison for these methods would be with $\text{AdvEmbed}$ from Tables 1 and 2, rather than the $\text{Vanilla}$ baseline presented in Table 5.
>
> For your convenience, we have provided $\Delta_{\tt avg}$ scores for the methods from the paper in the table below.
>
> | Methods \ Tasks | Sentiment Classification | Entailment |
> |:--------|:------------:|:----:|
> | $\text{AdvEmbed}$ | $13.70$ | $\underline{5.72}$ |
> | $\text{Min}$  | $7.26$ | $0.44$ |
> | $\text{Rand}$ | $\underline{14.88}$ | $1.67$ |
> | $\text{RoAST}$  | $\bf 18.39$ | $\bf 7.63$ |
>
> From the table, one can observe that:
> - $\text{Min}$ significantly lags behind the $\text{AdvEmbed}$ baseline, indicating the ineffectiveness of updating only the less important parameters.
> - $\text{Rand}$ shows a slight improvement in sentiment classification, possibly due to the additional benefit of sparse updates in a task that is more susceptible to overfitting. This is suggested by the higher validation performance overall, despite fewer training samples for sentiment classification.
> - There is a clear pattern of improvement with selective updates as the importance of the selected parameters increases, as demonstrated by the order: $\text{Min}$ < $\text{Rand}$ < $\text{RoAST (Max)}$.
>
> Overall, these results clearly reveal the importance of selecting parameters based on their importance scores. We will make this clearer in our revised manuscript.
>
> ---
>
> #### **[Q]. Can you show some results of FreeLB++, which runs adversarial perturbation for 10 steps instead of one?**
> **[A].** Following your suggestion, we conducted additional experiments with FreeLB++ on the sentiment classification task, which runs adversarial perturbation for 10 steps without constraints of norm-bounded projection [1]. We also tuned the adversarial step size appropriately as the number of steps changed. The results are summarized in the below table.
>
> | Method  | Acc$_{\tt in}$ | Acc$_{\tt shift}$ | Acc$_{\tt adv}$ | ECE ($\downarrow$) | AUROC | $\Delta_{\tt avg}$ |
> |:--------|:------------:|:----:|:------------:|:------------:|:----:|:------------:|
> | Vanilla | $96.29 $ | $91.79 $ | $ 66.30 $ | $7.11 $ | $86.72 $ | $0.00$ |
> | FreeLB | $\underline{96.33}$ | $\underline{91.94}$ | $ 70.07 $ | $6.49 $ | $89.82 $ | $9.21$ |
> | FreeLB++ | $96.14 $ | $91.79$ | $\underline{72.45}$ | $\bf 5.44$ | $\bf 90.79$ | $\underline{11.02} $ |
> | RoAST (Ours) | $\bf 96.87$ | $\bf 92.38$ | $\bf 72.57 $ | $\underline{5.45}$ | $\underline{90.37}$ | $\bf 18.39$ |
>
> Here, one can observe that $\text{FreeLB++}$ outperforms $\text{FreeLB}$ with a large gap, especially in $\text{ACC}\_{\tt adv}$ (70.07 $\rightarrow$ 72.45). Consequently, $\text{FreeLB++}$ is better than $\text{FreeLB}$ for multi-perspective robustness as well ($\Delta\_{\tt avg}: 9.21 \rightarrow 11.02$). Since $\text{RoAST}$ also uses a single step for constructing adversarial perturbation, these results indicate a potential room for further improvement in our method at the additional cost of the increased number of steps. We will include the respective discussions and results in the final draft.
>
> ---
>
> We hope this response will clarify your concerns. Please let us know if there are any unclear points remaining.
>
> Sincerely,
> Authors.
>
> [1] Li et al, Searching for an Effective Defender: Benchmarking Defense Against Adversarial Word Substitution, EMNLP 2021

---

### Official Review · Reviewer_C6Wu · 2023-08-02

**Soundness:** 2

**Excitement:**

2: Mediocre: This paper makes marginal contributions (vs non-contemporaneous work), so I would rather not see it in the conference.

**Missing References:**

Some related works are missing. List a few:

CreAT
Wu et al. Toward adversarial training on contextualized language representation

TRADES
Zhang et al. Theoretically principled trade-off between robustness and accuracy

AWP
Wu et al. Adversarial weight perturbation helps robust generalization


**Paper Topic And Main Contributions:**

This paper proposes a simple fine-tuning technique Robustifying LMs via Adversarial perturbation with Selective Training (RoAST) that aims to improve the multi-perspective robustness of LMs. To be specific, RoAST introduce adversarial embedding-level continuous perturbation during finetuning while the model parameters are selectively updated upon the square of their gradients. They conduct experiments on six different types of LMs on sentiment classification and entailment tasks.

**Questions For The Authors:**

What does the ‘coined’ mean in line 70?

Why is the matric of the experiment use a relative average form? Except for ECE, what is the advantage of it compared to a directly average form?

Why does the score of accuracy and AUROC in the same form in every table but in line 368-369 it says the best score of each metric: 100 for accuracy and 1 for AUROC?

Experiments using different LMs aim to demonstrate the effectiveness of RoAST, rather than compare different LMs. Why using the vanilla fine-tuned BERT-large as the baseline to calculate the average relative improvement in the experiment of other LMs rather than use the corresponding LMs score? Does this change of the matric make no difference?


**Reasons To Accept:**

1. The collaborative framework pursuing unified robustness has been less explored and this paper incorporates two important sources: robustness on the perturbed inputs and generalizable knowledge in pre-trained LMs and thus improve the robustness of LMs in multiple perspective.

2. This paper conducts extensive experiments and have best weighted average performance across the four robustness aspects along with a standard validation accuracy compared to state-of-the-art finetuning methods.


**Reasons To Reject:**

General weakness:

1. This paper only incorporates one kind of perturbed inputs techniques and one kind of preserving generalizable knowledge techniques, and thus far from a general framework to incorporate two kinds of techniques. The paper formalizes and derives that under mild assumptions, the masked gradient serves as an unbiased estimate of the original gradient, and the norm of covariance is upper bounded. But the derivation seems to be not very clear. Why is $g^{(i)}$ assumed to be in the form presented in line 1423? How to derive from line 1428 to line 1429? Meanwhile, it should be explained what implications this corollary has, such as whether it indicates that gradient masking does not affect the convergence of the model, etc.

2. The selective training method this paper propose is simple and intuitive, and distinct methods for each perspective of robustness are available. The overall paper lacks novelty.

3. The experimental results show that on entailment task, this paper doesn’t achieve best score on any single matric. Moreover, considering the ‘relative’ part in the average relative improvement $\Delta_{avg}$, this score is scaling up (for example, nearly ten times of $Acc_{in}$) compared to the original score and thus 2 points improvements compared to the best score of state-of-the-art finetuning method is not satisfiable. Although this paper includes numerous experiments, the analysis does not seem to match the workload. For instance, Table 2 does not show a significant improvement of ROAST compared to the baselines in the entailment task, but the authors have not analyzed the reason. Similarly, the analyses in the ablation study only describe the results in the tables, without providing sufficient reasoning or evidence for the effectiveness or shortcomings of ROAST.
Only BERT-based models are considered in the paper.
Decoder-only structure is more popular with the emergence of LLMs. It is promising to study the robustness on them.



**Reproducibility:**

3: Could reproduce the results with some difficulty. The settings of parameters are underspecified or subjectively determined; the training/evaluation data are not widely available.

**Reviewer Confidence:**

5: Positive that my evaluation is correct. I read the paper very carefully and I am very familiar with related work.

---

> ### Author Rebuttal · Authors · 2023-08-29
>
> We sincerely appreciate your efforts and insightful comments to improve the manuscript. We respond to each of your comments one-by-one in what follows. In tables, all the values are mean across 3 random seeds.
>
> ---
>
> #### **[R1-1]. This paper only incorporates one kind of perturbed inputs techniques and one kind of preserving generalizable knowledge techniques, and thus far from a general framework to incorporate two kinds of techniques.**
> **[A].** Thank you for pointing it out. We first clarify that our goal is NOT to create a general framework to incorporate two kinds of techniques. Rather, our goal is to establish a highly effective single framework for enhancing the multi-perspective robustness of LMs simultaneously. We believe that this is an important direction in practice, as it can mitigate the complexities and inefficiencies associated with exploring numerous combinations.
>
> Regarding this, we remark that our approach consistently surpasses existing methods that attempt to merge two kinds of techniques, as highlighted in Section 4.3. Through comparisons with prior works [1,2] and other advanced technique combinations aiming at the same intuition of incorporating, our results demonstrate the effectiveness of our method as a single framework to incorporate two kinds of techniques.
>
> ---
>
> #### **[R1-2]. The paper formalizes and derives that under mild assumptions, the masked gradient serves as an unbiased estimate of the original gradient, and the norm of covariance is upper bounded. But the derivation seems to be not very clear.**
> **[A].** Thank you for the insightful questions.
> - ***Why $g^(i)$ is assumed to be in the form presented in line 1423?***: Indeed, this form is based on the Central Limit Theorem (CLT); CLT suggests that, under mild assumptions, the distribution of the sum (or average) of a large number of independent, identically distributed variables will be approximately normal, regardless of the underlying distribution. In the context of our work, this would imply that the gradients, which are calculated in the mini-batch of the samples, would approximate a normal distribution. We additionally remark that this is a common simplifying assumption in the literature [3,4,5].
>
> - ***How to derive from line 1428 to line 1429?***: Line 1429 ($\mathbb{E}[\widetilde{g}] = \mathbb{E}[g]$) is derived from line 1428 ($\widetilde{g} := \big(m(\theta)/p(\theta)\big) \odot g$) as follow. We will add these details in the final draft:
> $\mathbb{E}[\widetilde{g}]=\mathbb{E}[( m(\theta) / p(\theta) ) \odot g]=\mathbb{E}[ m(\theta) \odot g]/p(\theta)=(\mathbb{E}[m(\theta)] \odot \mathbb{E}[g])/p(\theta)$ $  = (p(\theta) \times \frac{\partial \mathcal{L}}{\partial \theta}) / p(\theta)  = \frac{\partial \mathcal{L}}{\partial \theta}) = \mathbb{E}[g]$
>
> - ***Meanwhile, it should be explained what implications this corollary has, such as whether it indicates that gradient masking does not affect the convergence of the model, etc.***: The corollary establishes that, under certain conditions, the masked gradient $\widetilde{g}(\theta)$ is an unbiased estimator of the original gradient $g(\theta)$. The implication of this result is that the masked gradient correctly identifies the true gradient on average; therefore, the variance introduced by masking doesn't introduce a systematic bias that could deviate the model from converging to its optimal parameters.
>
>   In addition, the upper bound on the norm of covariance provides confidence in the stability of this estimator. In practical terms, it assures us that the masked gradient won't be too volatile or far off from the true gradient, thus safeguarding the convergence properties of the optimization process.
>
>   To clarify the implications of the corollary, we will incorporate this discussion in the final manuscript.
>
> ---
>
> #### **[R2]. The selective training method this paper propose is simple and intuitive, and distinct methods for each perspective of robustness are available. The overall paper lacks novelty.**
> **[A].** We would like to emphasize that this work is the first to explore the problem of multi-perspective robustness of LMs, with a novel approach for addressing this problem.
> - For reliable deployment to real-world applications, LMs must be robust across multiple dimensions simultaneously, rather than just a specific one. Despite this practical importance, the direction of multi-perspective robustness has yet to be explored. While each perspective of robustness for LMs has been separately explored with distinct methods (as we have also pointed out in the introduction), enhancing the multi-perspective robustness of LMs is still non-trivial; naively combining the methods would largely increase the computational cost and hyper-parameters and could even conflict with each other.
> - In addition to exploring a new problem, we propose a new training method, $\text{RoAST}$, which is simple and intuitive, as highlighted by you. As shown in the experimental results, $\text{RoAST}$ outperforms various approaches to enhance the robustness of LMs (including their combination) and is compatible with various types of LMs. Therefore, we expect that $\text{RoAST}$ can serve as a unified way to effectively enhance the multi-perspective robustness of LMs.
>
> Overall, we believe that both the introduction of a new problem of multi-perspective robustness of LMs and the design of a novel training method for it are valuable contributions to the community, as highlighted by Reviewer o3bN. We’ll make the contribution more clear during revision.
>
> ---
>
> #### **[R3-1]. The experimental results show that on entailment task, this paper doesn’t achieve best score on any single matric. Moreover, considering the ‘relative’ part in the average relative improvement $\Delta_{\tt avg}$, this score is scaling up (for example, nearly ten times of $\text{Acc}_{\tt in}$) compared to the original score and thus 2 points improvements compared to the best score of state-of-the-art finetuning method is not satisfiable.**
> **[A.]** Firstly, we would like to clarify that the primary goal of this work is to improve the multi-perspective robustness of LMs, NOT to improve a single metric of a specific perspective. Our approach did not achieve the best score on any single metric in the case of the entailment task, but it is more important to recognize that our method demonstrates the best results when considering multi-perspective robustness. Additionally, we remark that our method even achieves the best scores for three different metrics in the case of the sentiment classification task.
>
> Moreover, we use the average of relative improvement as it is more appropriate to measure the multi-perspective robustness than the average of absolute improvement, not to scale up the values (please see the detailed answer for this, in the response for Q2 below). However, we also recognize its weak points, such as the risk of amplification, which is the reason why we additionally report $\text{Rank}_{\tt avg}$ (as it does not have similar issues). We emphasize that $\text{RoAST}$ exhibits the lowest rank among the state-of-the-art fine-tuning methods on both sentiment classification and entailment tasks.
>
> Nevertheless, to address your concerns about this, we additionally calculate the absolute average improvement ($\text{Abs}\_{\tt avg}$) of {$\text{Acc}\_{\tt in}, \text{Acc}\_{\tt shift}, \text{Acc}\_{\tt adv}$, $100 - \text{ECE}$, $100 * \text{AUROC}$}, on the sentiment classification task. The results are presented in the below table.
>
> | Method  | Vanilla | WordDrop | R-Drop | HiddenCut | AdvWeight | AdvEmbed | FreeLB | SMART | RIFT | ChildPTune | R3F | WCons | LP-FT | RoAST (Ours) |
> |:--------|:------------:|:----:|:------------:|:------------:|:----:|:------------:|:------------:|:------------:|:----:|:------------:|:------------:|:----:|:------------:|:------------:|
> | $\text{Abs}_{\tt avg}$  | $76.47$ | $76.36$ | $76.96$ | $76.91$ | $76.94$ | $77.20$ | $77.13$ | $\underline{77.26}$ | $76.95$ | $74.72$ | $76.93$ | $77.05$ | $76.99$ | $\textbf{77.63}$ |
>
> Here, one can observe that our method still outperforms the state-of-the-art fine-tuning method with a large gap; $\text{RoAST}$ exhibits 46.72% relative improvement on $\text{Abs}_{\tt avg}$, compared to $\text{SMART}$ (1.16% vs 0.79%). This result further demonstrates the effectiveness of our method, and we do believe that our empirical results clearly show the merit of the proposed framework (as highlighted by reviewers Gq2X and o3bN). We will add this result and related discussion in the revised paper.
>
> ---
>
> #### **[R3-2]. Although this paper includes numerous experiments, the analysis does not seem to match the workload. For instance, Table 2 does not show a significant improvement of ROAST compared to the baselines in the entailment task, but the authors have not analyzed the reason. Similarly, the analyses in the ablation study only describe the results in the tables, without providing sufficient reasoning or evidence for the effectiveness or shortcomings of ROAST.**
>
> **[A].** Thank you for the suggestion. We would like to first note that we presented various analyses of $\text{RoAST}$ by focusing on designing appropriate experiments, such as measuring the distance to pre-trained LM or the importance of selection strategies, in Section 4.3. However, we agree that there is room for improvement by providing additional intuitions of the results. In the revision, we will incorporate more analysis of the results, especially for the following aspects as suggested.
>
> ***Reasoning for different effectiveness between tasks.*** We clarify that the different effectiveness for each task is commonly observed, not only for $\text{RoAST}$. Overall, advanced fine-tuning methods are more effective in the sentiment classification task than in the entailment task. One possible reason is the difference in the intrinsic difficulty of the task and training dataset size. While the size of the training data is much smaller for SST-2 (67k vs MNLI: 393k), the accuracy is much higher in the sentiment classification task. This indicates that the model could be more vulnerable to overfitting, and hence regularization of training could be more effective.
>
> ***More analyses in the ablation study.*** In addition to the presented ones (lines 496-514), we provide the following additional reasonings from the results of the ablation study in Table 4.
> - *Implications of performance degradation with $p(\theta)$ (c -> d)*: (c) Hard thresholding for only updating the important parameters with $s(\theta)$ can strictly preserve the original parameters of the pre-trained LM, which is less important to the given task, while (d) using $p(\theta)$ updates all the model parameters with the re-scaled gradients. Therefore, the performance decrease with $p(\theta)$ compared to $s(\theta)$ indicates the advantages of imposing sparsity rather than re-scaling to update the model for a given task while preserving the generalizable knowledge of the pre-trained LM.
> - *Implications of performance improvement with $m(\theta)$ (c,d -> $\text{RoAST}$)*: While (c) the selective training with hard thresholding showed the improvement, its effectiveness can be limited as it distorts the model update from the original direction through deterministic updates of important parameters. Compared to this, $\text{RoAST}$ can continuously take advantage of selective training while reducing such distortion by sampling and re-scaling the mask $m(\theta)$ from $p(\theta)$.  This advantage of $\text{RoAST}$ is verified by the improvements over (c,d).
>
> ---
>
> #### **[R3-3]. Only BERT-based models are considered in the paper. Decoder-only structure is more popular with the emergence of LLMs. It is promising to study the robustness on them.**
> **[A].** We agree that our approach has the potential to be applicable beyond BERT-based models, and demonstrating its effectiveness in broader types of models would strengthen our paper. In response to your suggestion, we conducted additional experiments on the sentiment classification task with GPT2-large, a popular decoder-only LM, to validate the applicability of our approach. Following [6], we added a linear classifier head on the last token’s embedding output for fine-tuning. The results are presented in the table below.
>
> | Method  | Acc$_{\tt in}$ | Acc$_{\tt shift}$ | Acc$_{\tt adv}$ | ECE ($\downarrow$) | AUROC | $\Delta_{\tt avg}$ |
> |:--------|:------------:|:----:|:------------:|:------------:|:----:|:------------:|
> | Vanilla | $\underline{94.99}$ | $\textbf{84.33}$ | $62.38$ | $\underline{8.01}$ | $95.30$ | $0.00$ |
> | AdvEmbed | $94.92$ | $\underline{84.19}$ | $\underline{63.58}$ | $\textbf{7.90}$ | $\underline{95.81}$ | $\underline{2.60}$ |
> | WCons | $94.88$ | $80.72$ | $61.80$ | $9.06$ | $93.57$ | $-15.37$ |
> | RoAST (Ours) | $\textbf{95.03}$ | $81.91$ | $\textbf{64.96}$ | $8.52$ | $\textbf{97.16}$ | $\textbf{5.07}$ |
>
> Here, we first observe that the improvements with baseline methods are largely limited. We speculate that this ineffectiveness may occur due to the different nature of decoder-only LMs compared to encoder-only ones, such as BERT, as it could result in different effectiveness of the baseline algorithms and tuned hyper-parameters, which were originally developed and demonstrated only using encoder-only models; for instance, most of the baselines [2,5,7,8] have been demonstrated under encoder-only models and not shown the results of decoder-only ones.
>
> Nevertheless, our $\text{RoAST}$ approach continues to enhance the multi-perspective robustness of GPT2-large, with an average relative improvement of 5.07% compared to the Vanilla method. These results indicate that the effectiveness of our approach is not limited to BERT-based LMs. Hence, we hope that our work could contribute to broader applications in the future. We will add more results and corresponding discussions to the final draft. Thank you for your suggestion to strengthen our paper.
>
> ---
>
> #### **[Q1] What does the ‘coined’ mean in line 70?**
> **[A].** The term ‘coined’ in line 70 refers to the act of introducing a new term or phrase for something, especially in an academic context. In our paper, the term "Robustifying LMs with Adversarial perturbation with Selective Training" is introduced as the full name of our proposed technique, which we then abbreviate as "$\text{RoAST}$" for ease of reference throughout the paper.
>
> ---
>
> #### **[Q2] Why is the matric of the experiment use a relative average form? Except for ECE, what is the advantage of it compared to a directly average form?**
> **[A].** We used a relative average instead of a direct average for measuring multi-perspective robustness, as it is more appropriate for the problem at hand. To evaluate LMs' multi-perspective robustness, it is crucial to equally incorporate multiple metrics from various perspectives. However, these metrics often have different scales, and the direct average is limited in preventing a single metric from dominating the aggregate results due to its numerical magnitude. For instance, the average absolute improvement in $\text{Acc}\_{\tt adv}$ is much larger than in $\text{Acc}\_{\tt in}$ due to the former's relatively small values. In contrast, the relative average addresses this issue by considering normalized metrics. We'll make sure to explain this more clearly in our final manuscript.
>
> ---
>
> #### **[Q3] Why does the score of accuracy and AUROC in the same form in every table but in line 368-369 it says the best score of each metric: 100 for accuracy and 1 for AUROC?**
> **[A].** Many thanks for the careful reading, particularly in highlighting the inconsistency between the tables and the text in lines 368-369. In our tables, we have presented the value as 100 × $\text{AUROC}$ instead of $\text{AUROC}$, for a better presentation by aligning its scale (including significant figures) with other metrics such as accuracy and $\text{ECE}$. However, we remark that the metrics for comparison ($\Delta_{\tt avg}$ and $\text{Rank}_{\tt avg}$) remain unchanged and are unaffected by this re-scaling. To enhance clarity and prevent any confusion, we will integrate the following statement into the experimental setup in the final draft. Thank you again for pointing out this!:
> “We present the AUROC values as 100 × AUROC in our tables to maintain a consistent scale with other metrics.”
>
> ---
>
> #### **[Q4] Experiments using different LMs aim to demonstrate the effectiveness of RoAST, rather than compare different LMs. Why using the vanilla fine-tuned BERT-large as the baseline to calculate the average relative improvement in the experiment of other LMs rather than use the corresponding LMs score? Does this change of the matric make no difference?**
> **[A].** Thank you for the insightful comments. We use BERT-large as a universal baseline to calculate relative improvement across different LMs to facilitate comparison in terms of multi-perspective robustness. This choice was made to provide additional insight into the question of *“Which LM is the most robust”*. While answering this question is important, we also acknowledge that using the corresponding LM's score as the baseline could provide additional insights into how $\text{RoAST}$ performs with each specific LM.
> Hence, to address your concerns about the potential impact of this choice on the metrics, we recalculate Table 3 with the corresponding LM's score and present it below. One can observe that $\text{RoAST}$ mostly outperforms the baselines except in only 1 case, and achieves large improvements compared to baselines in both tasks. This result clearly demonstrates the robustness and effectiveness of $\text{RoAST}$ with respect to different LMs. We will add the corresponding discussions and results in the final draft.
>
> | Sentiment Classification  | Vanilla | AdvEmbed | WCons | RoAST |
> |:--------|:------------:|:----:|:------------:|:------------:|
> | BERT-large | $0.00$ | $\underline{18.24}$ | $15.85$ | $\textbf{22.76}$ |
> | RoBERTa-large | $0.00$ | $13.70$ | $\underline{15.47}$ | $\textbf{18.39}$ |
> | ALBERT-xxlarge | $0.00$ | $\underline{21.34}$ | $18.25$ | $\textbf{30.62}$ |
> | XLNet-large | $0.00$ | $\underline{1.44}$ | $-1.76$ | $\textbf{12.74}$ |
> | ELECTRA-large | $0.00$ | $\underline{6.44}$ | $-2.83$ | $\textbf{8.29}$ |
> | DeBERTa-large | $0.00$ | $\underline{10.53}$ | $4.49$ | $\textbf{20.62}$ |
> | Average | $0.00$ | $\underline{11.95}$ | $8.25$ | $\textbf{18.90}$ |
>
> | Entailment  | Vanilla | AdvEmbed | WCons | RoAST |
> |:--------|:------------:|:----:|:------------:|:------------:|
> | BERT-large | $0.00$ | $\underline{22.85}$ | $20.23$ | $\textbf{25.34}$ |
> | RoBERTa-large | $0.00$ | $\underline{5.72}$ | $2.99$ | $\textbf{7.63}$ |
> | ALBERT-xxlarge | $0.00$ | $\underline{2.60}$ | $-3.83$ | $\textbf{8.19}$ |
> | XLNet-large | $0.00$ | $2.44$ | $\underline{2.75}$ | $\textbf{3.30}$ |
> | ELECTRA-large | $0.00$ | $\textbf{7.89}$ | $-0.70$ | $\underline{4.39}$ |
> | DeBERTa-large | $0.00$ | $2.81$ | $\underline{3.79}$ | $\textbf{11.93}$ |
> | Average | $0.00$ | $\underline{7.39}$ | $4.21$ | $\textbf{10.13}$ |
>
> ---
>
> #### **[Missing References] Some related works are missing.**
> **[A].** Thank you for letting us know about the additional related works! As the line of work for the adversarial robustness, we agree that their inclusion will significantly enhance the comprehensiveness and depth of our related works section. We will include them in the related work section of our final manuscript.
>
> ---
> We hope this response will clarify your concerns. Please let us know if there are any unclear points remaining.
>
> Sincerely,
> Authors.
>
> [1] Dong et al., How Should Pre-Trained Language Models Be Fine-Tuned Towards Adversarial Robustness?., NeurIPS 2021
> [2] Jiang et al., SMART: Robust and Efficient Fine-Tuning for Pre-trained Natural Language Models through Principled Regularized Optimization., ACL 2020
> [3] Smith and Le., A Bayesian Perspective on Generalization and Stochastic Gradient Descent., ICLR 2018
> [4] Zhang et al., Which Algorithmic Choices Matter at Which Batch Sizes? Insights From a Noisy Quadratic Model., NeurIPS 2019
> [5] Xu et al., Raise a Child in Large Language Model: Towards Effective and Generalizable Fine-tuning., EMNLP 2021
> [6] Radford et al., Improving Language Understanding by Generative Pre-Training., OpenAI
> [7] Zhu et al., FreeLB: Enhanced Adversarial Training for Natural Language Understanding., ICLR 2020
> [8] Chen et al., Recall and Learn: Fine-tuning Deep Pretrained Language Models with Less Forgetting., EMNLP 2020

---

### Official Review · Reviewer_o3bN · 2023-08-04

**Soundness:** 4

**Excitement:**

3: Ambivalent: It has merits (e.g., it reports state-of-the-art results, the idea is nice), but there are key weaknesses (e.g., it describes incremental work), and it can significantly benefit from another round of revision. However, I won't object to accepting it if my co-reviewers champion it.

**Paper Topic And Main Contributions:**

The paper is addressing the problem of lack of robustness in language models. Even though it is a well studied problem, the authors present limitations of previous work and motivate their work as an exploration of a single effective way for fine-tuning LMs to improve its robustness in multiple perspectives. They propose a fine-tuning method to address this problem, namely Robustifying LMs with Adversarial perturbation with Selective Training (ROAST). The method is based on the idea of combining robustness on the perturbed input and generalizable knowledge learned during pre-training of LMs, during the fine-tuning stage. The authors show the superiority of their approach compared to various baselines in sentiment analysis and entailment tasks.

**Questions For The Authors:**

- Could the authors discuss the choice of embedding-level perturbation? Would the method also work for discrete token-level adversarial perturbations?


**Reasons To Accept:**

- The paper is well written, well motivated and easy to follow.
- The proposed approach, RoAST, is novel given that is adds a multi-perspective robustness of LMs in a unified way.
- The experimental setup is solid, with multiple baselines and thorough analysis (and ablation), making the results and conclusions strong.

**Reasons To Reject:**

- The main weakness of the paper is the limited experimentation in the tasks/datasets side. Even though the baselines and models used are thorough, admittedly the authors could have used more datasets from each tasks and more challenging benchmarks.

**Reproducibility:**

3: Could reproduce the results with some difficulty. The settings of parameters are underspecified or subjectively determined; the training/evaluation data are not widely available.

**Reviewer Confidence:**

3: Pretty sure, but there's a chance I missed something. Although I have a good feel for this area in general, I did not carefully check the paper's details, e.g., the math, experimental design, or novelty.

---

> ### Author Rebuttal · Authors · 2023-08-29
>
> We sincerely appreciate your efforts and insightful comments to improve the manuscript. We respond to each of your comments one-by-one in what follows. In tables, all the values are mean across 3 random seeds.
>
> ---
>
> #### **[R]. The main weakness of the paper is the limited experimentation in the tasks/datasets side. Even though the baselines and models used are thorough, admittedly the authors could have used more datasets from each tasks and more challenging benchmarks.**
> **[A].** We appreciate your recognition of the thoroughness of our baselines and models and thank you for the great suggestion! We agree that incorporating more datasets and challenging benchmarks would further strengthen our experiments.
>
> However, we would like to note that a significant number of diverse datasets were already included in our experiments to evaluate multi-perspective robustness on each task. For example, we used 16 and 22 datasets for sentiment classification and entailment tasks, respectively. Moreover, constructing a benchmark to evaluate multi-perspective robustness for a new task is not a trivial task, as it involves identifying and integrating separately explored datasets for each perspective. Nevertheless, we believe the additional results on more challenging tasks would greatly strengthen our paper. We will continue to work and supplement more results in the final draft.
>
> Overall, we believe that our experimentation serves as a useful contribution and a foundation for pursuing a unified approach to evaluate and enhance the multi-perspective robustness of LMs. We also hope that our benchmarks will continue to be extended with new datasets and tasks by the community, and serve as a meaningful step towards the more reliable deployment of LMs.
>
> ---
>
> #### **[Q]. Could the authors discuss the choice of embedding-level perturbation? Would the method also work for discrete token-level adversarial perturbations?**
>
> **[A].** We chose embedding-level perturbation over token-level one, as it is more computationally efficient and better suited for enhancing multi-perspective robustness.
>
> - Construction of token-level adversarial perturbations requires more computation to solve the discrete optimization problem, and additional regularization is often introduced to prevent degenerate cases such as significant changes in semantic or lexical violation [1,2,3]. For example, a relatively simple construction of token-level perturbation with [3] requires 15% more times per iteration, compared to embedding-level perturbation.
> - While the token-level adversarial perturbation is effective for adversarial robustness, it often comes at the cost of a decrease in clean accuracy [4,5] due to the relatively large perturbation on a discrete space. In contrast, the embedding-level perturbation can be constructed by adding small noise to continuous space, making it more feasible to train the model without the loss of accuracy [6,7]
>
> Moreover, to address your comments, we conduct additional experiments to validate the effectiveness of $\text{RoAST}$ with discrete token-level adversarial perturbations by fine-tuning RoBERTa-large using $\text{VAT-D}$ [3]. We used the same hyper-parameters for discrete token-level perturbation as in [3], and for $\text{RoAST}$, we used the same values previously found with embedding-level perturbation. The results are shown in the table below.
>
> | Method  | Acc$_{\tt in}$ | Acc$_{\tt shift}$ | Acc$_{\tt adv}$ | ECE ($\downarrow$) | AUROC | $\Delta_{\tt avg}$ |
> |:--------|:------------:|:----:|:------------:|:------------:|:----:|:------------:|
> | $\text{Vanilla}$ | $\textbf{96.29}$ | $\textbf{91.79}$ | $66.30$ | $7.11$ | $86.72$ | $0.00$ |
> | $\text{VAT-D}$ | $95.83$ | $\underline{90.86}$ | $\underline{70.37}$ | $\textbf{6.33}$ | $\underline{90.39}$ | $\underline{5.37}$ |
> | $\text{VAT-D + RoAST}$ | $\underline{96.27}$ | $90.45$ | $\textbf{72.12}$ | $\underline{6.94}$ | $\textbf{92.16}$ | $\textbf{8.73}$ |
>
> Here, we observe that the discrete token-level adversarial perturbation can improve the multi-perspective robustness, especially for adversarial robustness ($\text{Acc}\_{\tt adv}$). However, it comes at the cost of a degradation in clean accuracy ($\text{Acc}\_{\tt in}$). With $\text{RoAST}$, such degradation can be mitigated and the overall robustness of the model could be improved. We will include the respective discussions and results in the final draft.
>
> ---
>
> Please let us know if there are any more comments; we would be more than happy to continue the discussion!
>
> Sincerely,
> Authors.
>
> [1] Jin et al., Is BERT Really Robust? A Strong Baseline for Natural Language Attack on Text Classification and Entailment., AAAI 2020
> [2] Li et al., BERT-ATTACK: Adversarial Attack Against BERT Using BERT., EMNLP 2020
> [3] Park et al., Consistency Training with Virtual Adversarial Discrete Perturbation., NAACL 2022
> [4] Jia et al., Certified Robustness to Adversarial Word Substitutions., EMNLP 2019
> [5] Dong et al., Towards Robustness Against Natural Language Word Substitutions., ICLR 2021
> [6] Zhu et al., FreeLB: Enhanced Adversarial Training for Natural Language Understanding., ICLR 2020
> [7] Jiang et al., SMART: Robust and Efficient Fine-Tuning for Pre-trained Natural Language Models through Principled Regularized Optimization., ACL 2020

---

### Meta-Review · Area_Chair_Bqby · 2023-09-25

**Recommendation:** 3

**Metareview:**

This paper presents an easy-to-follow approach called RoAST, which aims to improve the robustness of language models (LMs) from multiple perspectives.
This approach is unique as it considers both robustness on perturbed inputs and generalizable knowledge in pre-trained LMs.
The experimental setup is robust, including multiple baselines and thorough analysis, strengthening the results and conclusions. The paper's extensive experiments show the best weighted average performance in four robustness aspects, alongside a standard validation accuracy compared to other finetuning methods. Overall, RoAST has the potential to become a more robust and more generalized technique for various NLP tasks.

From the reviewers, this paper only showed experiments using limited tasks and datasets, thus recommending the use of challenging benchmarks and non-BERT-based models like decoder-only structures.
Overall, the paper is deemed to require further analysis to explain the difference in approaches, which could be more exciting to the community.

---

### Decision · Program_Chairs · 2023-10-07

**Decision:**

Accept-Findings

**Comment:**

This paper presents an easy-to-follow approach called RoAST, which aims to improve the robustness of language models (LMs) from multiple perspectives.
This approach is unique as it considers both robustness on perturbed inputs and generalizable knowledge in pre-trained LMs.
The experimental setup is robust, including multiple baselines and thorough analysis, strengthening the results and conclusions. The paper's extensive experiments show the best weighted average performance in four robustness aspects, alongside a standard validation accuracy compared to other finetuning methods. Overall, RoAST has the potential to become a more robust and more generalized technique for various NLP tasks.

From the reviewers, this paper only showed experiments using limited tasks and datasets, thus recommending the use of challenging benchmarks and non-BERT-based models like decoder-only structures.
Overall, the paper is deemed to require further analysis to explain the difference in approaches, which could be more exciting to the community.